# Virological characteristics of the SARS-CoV-2 Omicron XBB.1.5 variant

Tomokazu Tamura [1,2,3,4,5,6,41], Takashi Irie [7,41], Sayaka Deguchi[8,41], Hisano Yajima [9,41], Masumi Tsuda [10,11,41], Hesham Nasser [12,13,41], Keita Mizuma[14,41], Arnon Plianchaisuk [15,41], Saori Suzuki [1,2,3,5,41], Keiya Uriu[15,16], Mst Monira Begum[12], Ryo Shimizu[12], Michael Jonathan [12], Rigel Suzuki[1,2,3,5], Takashi Kondo[3], Hayato Ito[2], Akifumi Kamiyama[2], Kumiko Yoshimatsu [17], Maya Shofa[18,19], Rina Hashimoto [8], Yuki Anraku [20], Kanako Terakado Kimura[9], Shunsuke Kita [20], Jiei Sasaki[9], Kaori Sasaki-Tabata[21], Katsumi Maenaka [6,20,22,23], Naganori Nao[5,6,24], Lei Wang [10,11], Yoshitaka Oda[10], The Genotype to Phenotype Japan (G2P-Japan) Consortium*, Terumasa Ikeda [12], Akatsuki Saito [18,19,25], Keita Matsuno [5,6,14,26], Jumpei Ito [15,27], Shinya Tanaka [10,11]✉, Kei Sato [15,16,27,28,29,30,31]✉, Takao Hashiguchi [9,28,32]✉, Kazuo Takayama [8,33]✉ & Takasuke Fukuhara [1,2,3,4,5,6,32,34]✉

Circulation of SARS-CoV-2 Omicron XBB has resulted in the emergence of XBB.1.5, a new Variant of Interest. Our phylogenetic analysis suggests that XBB.1.5 evolved from XBB.1 by acquiring the S486P spike (S) mutation, subsequent to the acquisition of a nonsense mutation in ORF8. Neutralization assays showed similar abilities of immune escape between XBB.1.5 and XBB.1. We determine the structural basis for the interaction between human ACE2 and the S protein of XBB.1.5, showing similar overall structures between the S proteins of XBB.1 and XBB.1.5. We provide the intrinsic pathogenicity of XBB.1 and XBB.1.5 in hamsters. Importantly, we find that the ORF8 nonsense mutation of XBB.1.5 resulted in impairment of MHC suppression. In vivo experiments using recombinant viruses reveal that the XBB.1.5 mutations are involved with reduced virulence of XBB.1.5. Together, our study identifies the two viral functions defined the difference between XBB.1 and XBB.1.5.

Severe acute respiratory syndrome coronavirus 2 (SARS-CoV-2), the causative agent of the coronavirus disease 2019 (COVID-19), continues to circulate among humans, leading to the emergence of new variants. Our series of studies aiming to characterize SARS-CoV-2 variants[1–6] revealed that SARS-CoV-2 has enhanced intrinsic pathogenicity from the first identified B.1.1 variant to B.1.617.2 (Delta variant). Because of the spreading of SARS-CoV-2 and human vaccination leading to the acquisition of immunoprophylaxis, Omicron subvariants have begun

to evolve toward decreased intrinsic pathogenicity, increased transmissibility, and enhanced immune escape compared with ancestral variants. Since the resumption of economic activities and traveling, the Omicron variant has given rise to numerous subvariants across the globe in 2022[7]. Now, the XBB lineage, which emerged as a recombinant lineage between the second-generation BA.2 variants BJ.1 and BM.1.1.1 (BA.2.75.3.1.1.1; a descendant of BA.2.75), is outcompeting BQ.1.1, a descendant of BA.5, with continuous evolution. At the end of 2022, the

A full list of affiliations appears at the end of the paper. *A list of authors and their affiliations appears at the end of the paper.
✉e-mail: tanaka@med.hokudai.ac.jp; keisato@g.ecc.u-tokyo.ac.jp; hashiguchi.takao.1a@kyoto-u.ac.jp; kazuo.takayama@cira.kyoto-u.ac.jp; fukut@pop.med.hokudai.ac.jp

XBB.1.5 variant emerged, a descendant of XBB that acquired two substitutions in the spike (S) protein, S:G252V and S:F486P (S:S486P compared to XBB.1), and a translational stop in ORF8 (ORF8:G8stop). This variant is rapidly spreading in the USA, and is thus a Variant of Interest (VOI). To answer public concerns about whether XBB.1.5 could potentially spread across the globe, several groups, including us, have examined the immunoprophylactic ability against XBB.1.5 infection by monovalent or bivalent vaccination or infection with previous variants, and investigated epidemic dynamics analysis[6,8–15]. This suggested that the XBB.1.5 acquired S:S486P substitution which confers augmented ACE2 binding affinity without losing immune resistance leading to greater transmissibility compared to XBB.1. To the best of our knowledge, the comprehensive virological characteristics of XBB.1.5 are still unrevealed.

In this work, we used the indicated XBB subvariants (the XBB.1 TY41-795 and XBB.1.5 TKYmbc30523/2022 strains) clinically isolated from COVID-19 patients to investigate the virological characteristics in in vitro and in vivo models. Furthermore, to investigate the functional loss of ORF8 protein in XBB.1.5, we generated recombinant XBB.1 viruses bearing the S:S486P or ORF8:G8stop substitutions and defined the immunopathological effects in vivo.

## Results

### Evolutionary history of XBB.1.5

The SARS-CoV-2 XBB variant was first detected around August 2022[16], followed by its diversification into several descendant sublineages (Pango nomenclature; https://github.com/cov-lineages/pango-designation). XBB.1.5, an XBB subvariant, has predominated

worldwide since late February 2023 (Nextstrain, clade 23 A; https://nextstrain.org/ncov/gisaid/global/6m)[17]. XBB.1.5 exhibits a higher effective reproduction number than other XBB subvariants[6]. In this study, we traced the evolutionary history of XBB.1.5 through the reconstruction of a maximum likelihood-based phylogenetic tree using genomic sequences of SARS-CoV-2 isolates in the XBB lineage (Fig. 1, Supplementary Fig. 1, and Supplementary data 1). Regarding the difficulty of SARS-CoV-2 phylogenic analysis due to low supportive values[18], we reconstructed ten phylogenetic trees in total and compared their topology. All the trees suggest that XBB.1.5 emerged from an ancestor in the XBB.1 lineage. Compared with XBB, XBB.1.5 harbors S:G252V, S:F486P, and ORF8:G8stop mutations. To elucidate the occurrence order of these mutations, we reconstructed the ancestral genomic sequences and investigated the mutation occurrence along the phylogenetic trees (Fig. 1 and Supplementary Fig. 1). Our results from eight of the ten trees suggest that the S:G252V mutation putatively occurred first in an ancestor of XBB.1. Although most XBB.1 harbor the ORF8:G8stop mutation, this mutation occurred during the diversification of XBB.1 rather than in the most recent common ancestor (MRCA) of XBB.1. The S:S486P mutation occurred later in the putative MRCA of XBB.1.5, followed by the diversification of XBB.1.5, suggesting the contribution of the S:S486P mutation to the increased viral fitness of XBB.1.5 compared with XBB.1[6].

### XBB.1.5 immune escape

It has recently been reported that XBB.1.5, as well as XBB.1, exhibits profound resistance to neutralization mediated by the sera of BA.2 and BA.5 breakthrough infection[3,6]. To address whether XBB.1.5 is resistant to vaccine sera, we used three types of vaccine sera: fourth-dose monovalent sera, BA.1 bivalent vaccine sera, and BA.5 bivalent vaccine sera (Supplementary data 2). We showed that XBB.1.5 significantly evades fourth-dose monovalent vaccine sera (28.6-fold, $P < 0.0001$) (Fig. 2A), BA.1 bivalent vaccine sera (34.0-fold, $P < 0.0001$) (Fig. 2B), and BA.5 bivalent vaccine sera (16.8-fold, $P < 0.0001$) (Fig. 2C), more effectively than B.1.1. Similar to the observations on XBB.1[3], our results suggest that, regardless of the type of bivalent mRNA vaccine, XBB.1.5 is robustly resistant to vaccine sera.

### Structural characteristics of the XBB.1.5 S protein

To gain structural insights into the recognition of the ACE2 receptor and evasion from neutralizing antibodies by the XBB.1.5 S protein, we performed cryo-electron microscopy (cryo-EM) analysis of both the XBB.1.5 S ectodomain alone and the XBB.1.5 S–ACE2 complex (Fig. 3A–D, Supplementary Fig. 2, and Supplementary Table 1). Similar to BA.2.75 and XBB.1[2,3], the XBB.1.5 S ectodomain alone was reconstructed as two closed states (closed-1 and closed-2) at resolutions of 2.59 Å and 2.79 Å, respectively (Fig. 3A, Supplementary Fig. 2, and Supplementary Table 1). The receptor-binding domain (RBD) one-up state, frequently observed in the structure of the SARS-CoV-2 S protein, was rarely found in the XBB.1.5 S protein. Although the overall structures of the closed-1 and closed-2 states were essentially the same in both protomers and trimers between the S proteins of XBB.1 and XBB.1.5 (Fig. 3A), some differences were observed in the specific details. In the closed-1 state, a structural difference between the S proteins of XBB.1 and XBB.1.5 is the distinct loop structure consisting of L828 to Q836 within the fusion peptide (residues 816–855) (Fig. 3A). In addition, R567 and D571, which have no interaction in XBB.1, form a salt bridge in XBB1.5 (Fig. 3A). In the closed-2 state, a structural difference between the S proteins of XBB.1 and XBB.1.5 is that K811 interacts with S813 in XBB.1, whereas in XBB.1.5, K811 forms salt bridges with both D805 and D820 (Fig. 3A). A common feature shared by both the XBB.1 and XBB.1.5 S proteins is that the cryo-EM maps of the large part (residues 829–853) of the fusion peptide are highly disordered in both closed-2 states. Another common feature is that the closed-1 cryo-EM map showed higher resolution and well-packed RBDs, while

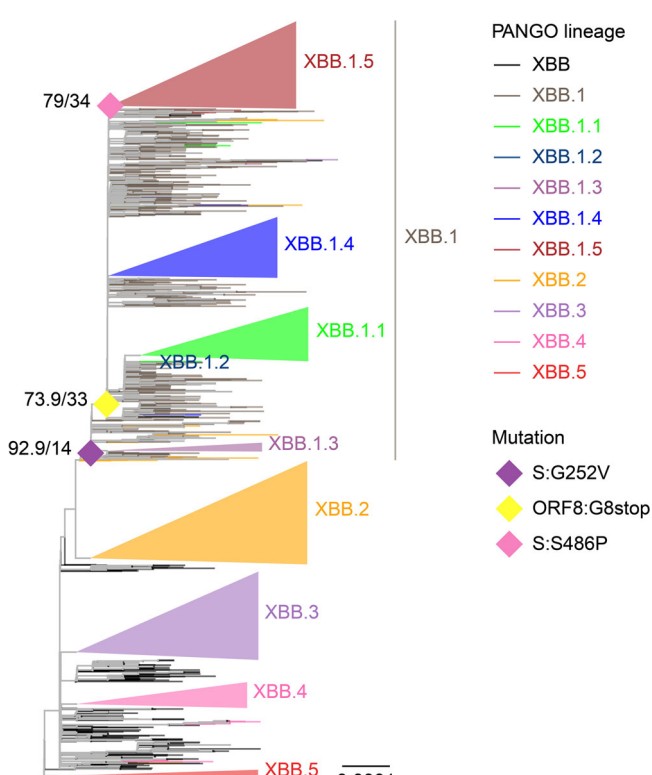

**Fig. 1 | Evolutionary history of the XBB.1.5 sublineage.** A representative maximum likelihood-based phylogenetic tree of SARS-CoV-2 in the XBB lineage. The XBB.1.4.1, XBB.3.1, and XBB.4.1 sublineages are included in the XBB.1.4, XBB.3, and XBB.4 lineages, respectively. Diamonds represent the occurrence of mutations of interest. Only mutation occurrences at internal nodes with at least 20 and also a half of descendant tips harboring the mutation are shown. Numbers at diamonds represent Shimodaira-Hasegawa-like approximate likelihood ratio test and ultrafast bootstrap supporting values, respectively.

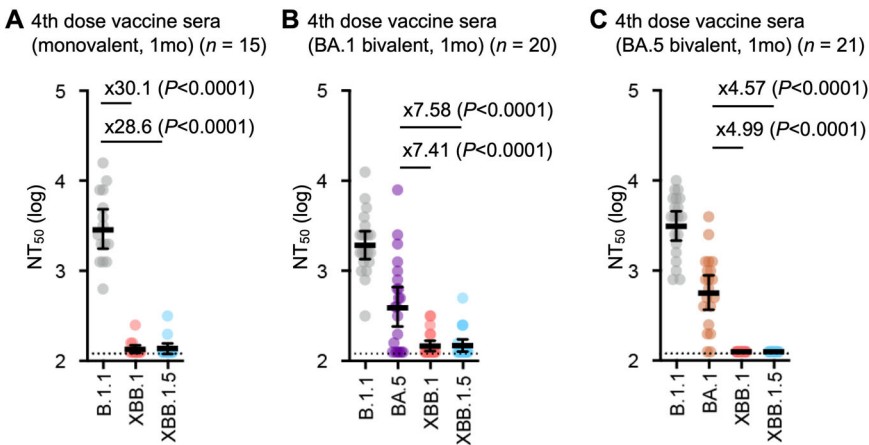

**Fig. 2 | Immune evasion of XBB.1.5.** Neutralization assays were performed with pseudoviruses harboring the S proteins of B.1.1, BA.1, BA.5, XBB.1, and XBB.1.5. The following sera were used: (**A**) fourth-dose vaccine sera collected one month after the fourth-dose monovalent vaccine (15 donors), (**B**) BA.1 bivalent vaccine (20 donors), and (**C**) BA.5 bivalent vaccine (21 donors). Assays for each serum sample were performed in triplicate to determine the 50% neutralization titer ($NT_{50}$). Each dot represents one $NT_{50}$ value, and the geometric mean and 95% CI are shown. Statistically significant differences were determined by two-sided Wilcoxon signed-rank tests. The $P$ values versus B.1.1 are indicated in the panels. The horizontal dashed line indicates the detection limit (120-fold). Red numbers indicate decreased and increased $NT_{50}$s. Information on the convalescent donors is summarized in Supplementary data 2.

the closed-2 cryo-EM map showed relatively lower resolution in the RBDs than in the whole map (Fig. 3A and Supplementary Fig. 2), suggesting that the RBDs are mobile.

The XBB.1.5 S–ACE2 complex structure was also reconstructed by cryo-EM analysis in both RBD one-up and two-up conformations, with resolutions of 3.04 Å and 3.20 Å, respectively (Fig. 3B, Supplementary Fig. 2, and Supplementary Table 1). Local refinement was performed on the RBD–ACE2 complex to observe the interaction between RBD and ACE2 with better resolution, and the structure was reconstructed at a resolution of 3.38 Å (Fig. 3B, Supplementary Fig. 2, and Supplementary Table 1). Although the S486P substitution occurs in the ACE2 binding motif of the XBB.1.5 S protein, the overall binding mode with ACE2 was not significantly changed compared with XBB.1. However, differences were observed in the RBD–ACE2 interface between XBB.1 and XBB.1.5 (Fig. 3C): in XBB.1.5, RBD Q493 interacts with both K31 and H34 of ACE2, whereas in XBB.1, RBD Q493 interacts only with K31 or H34 of ACE2 through an alternative conformation. In XBB.1, RBD Y453 interacts with ACE2 H34, but in XBB.1.5, the distance between them is more than 4.0 Å. A common feature shared by ACE2 receptors bound to the XBB.1 or XBB.1.5 S protein is that the N103-linked glycan of ACE2 is positioned towards the active site inside the ACE2 structure (Fig. 3C).

Finally, to understand the effects of the S486P substitution in the XBB.1.5 S protein, the RBD Y473–P491 loop regions in the closed-1, closed-2, and RBD–ACE2 structures were compared between XBB.1 and XBB.1.5 (Fig. 3D). The results show that the RBD Y473–P491 loop structures are similar in closed-1 or the RBD–ACE2 complex, while they are slightly shifted outward in closed-2 of XBB.1.5 (N487 interacts with S486 in XBB1, whereas in XBB1.5, N487 interacts with K478 (Fig. 3D).

### Fusogenicity of the XBB.1.5 S protein
To quantitatively monitor S protein-mediated fusion activity, we utilized DSP (dual split protein). DSP is composed of $DSP_{1-7}$ and $DSP_{8-11}$, which is a hybrid protein constituted by split *Renilla* luciferase (RL) and split green fluorescence protein (GFP)[19]. When $DSP_{1-7}$ and $DSP_{8-11}$ are associated after fusion, the reconstituted split proteins produce luminescence and fluorescence. Therefore, the fusogenicity of XBB.1.5 S was measured by the SARS-CoV-2 S-based fusion assay[1–4] using Calu-3 cells stably expressing $DSP_{1-7}$. The surface expression level of the XBB.1.5 S protein was comparable to those of BA.2 S and XBB.1 S (Fig. 4A and Supplementary Fig. 3A). As expected, the fusogenic ability of Delta S was greatest among we examined

(Supplementary Fig. 3A). The fusogenicity of XBB.1 S was greater than that of BA.2 S (Fig. 4B), which is consistent with our recent studies[3,6]. Similarly, the XBB.1.5 S protein was significantly more fusogenic than BA.2 S (Fig. 4B). However, there was no significant difference in fusogenicity between the XBB.1 and XBB.1.5 S proteins (Fig. 4B). These results indicate that XBB.1.5 S maintains comparable fusogenicity to XBB.1 S.

### Growth kinetics of XBB.1.5 in vitro
To investigate the growth kinetics of XBB.1.5 in in vitro cell-culture systems, we inoculated clinical isolates of Delta, XBB.1, and XBB.1.5 into multiple cell cultures. We quantified viral RNA copies in supernatants (Fig. 4C-F) as well as viral infectious titers (Supplementary Fig. 3B–D). At a multiplicity of infection (MOI) of 0.01, the multistep growth of Delta in Vero cells (Fig. 4C) and VeroE6/TMPRSS2 cells (Fig. 4D) was greater than that of XBB.1 and XBB.1.5, while the growth curves of XBB.1 and XBB.1.5 were almost comparable. At an MOI of 0.1 in VeroE6/TMPRSS2 cells, the growth curves of each virus became almost comparable to each other (Fig. 4E), probably due to the rapid spread of infection leading to quick killing of the cell cultures (Fig. 4F). As we previously reported, at 48 and 72 hours post-infection (h.p.i.), large multinuclear syncytia were readily observed in the Delta-infected cells, especially in the VeroE6/TMPRSS2 cells. However, few and small syncytia were observed in the cells infected with XBB.1 and XBB.1.5 (Fig. 4F). Compared with XBB.1 and XBB.1.5, the cytopathic effect (CPE) observed in XBB.1.5 occurred more quickly and severely than that in XBB.1, consistent with the slightly higher growth of XBB.1.5. We also quantified plaque size upon viral infection in Vero and VeroE6/TMPRSS2 cells (Supplementary Fig. 3E). While the size of plaques induced by XBB.1 and XBB.1.5 was comparable in VeroE6/TMPRSS2 cells, XBB.1.5 induced significantly larger plaques in Vero cells compared with XBB.1. This finding was further supported by the Viral ToxGlo assay where we observed a significantly higher cytopathic effects induced by XBB.1.5 than those induced by XBB.1 in Vero cells (Supplementary Fig. 3F). To assess the effects of XBB.1.5 infection on the airway epithelial and endothelial barriers, we employed an airway-on-a-chip system[3,20,21]. By assessing the quantity of virus that infiltrates from the top channel (Fig. 4G, left, Supplementary Fig. 3G) to the bottom channel (Fig. 4G, middle, Supplementary Fig. 3G), we can evaluate the capacity of viruses to breach the airway epithelial and endothelial barriers. Notably, the percentage of virus that infiltrated

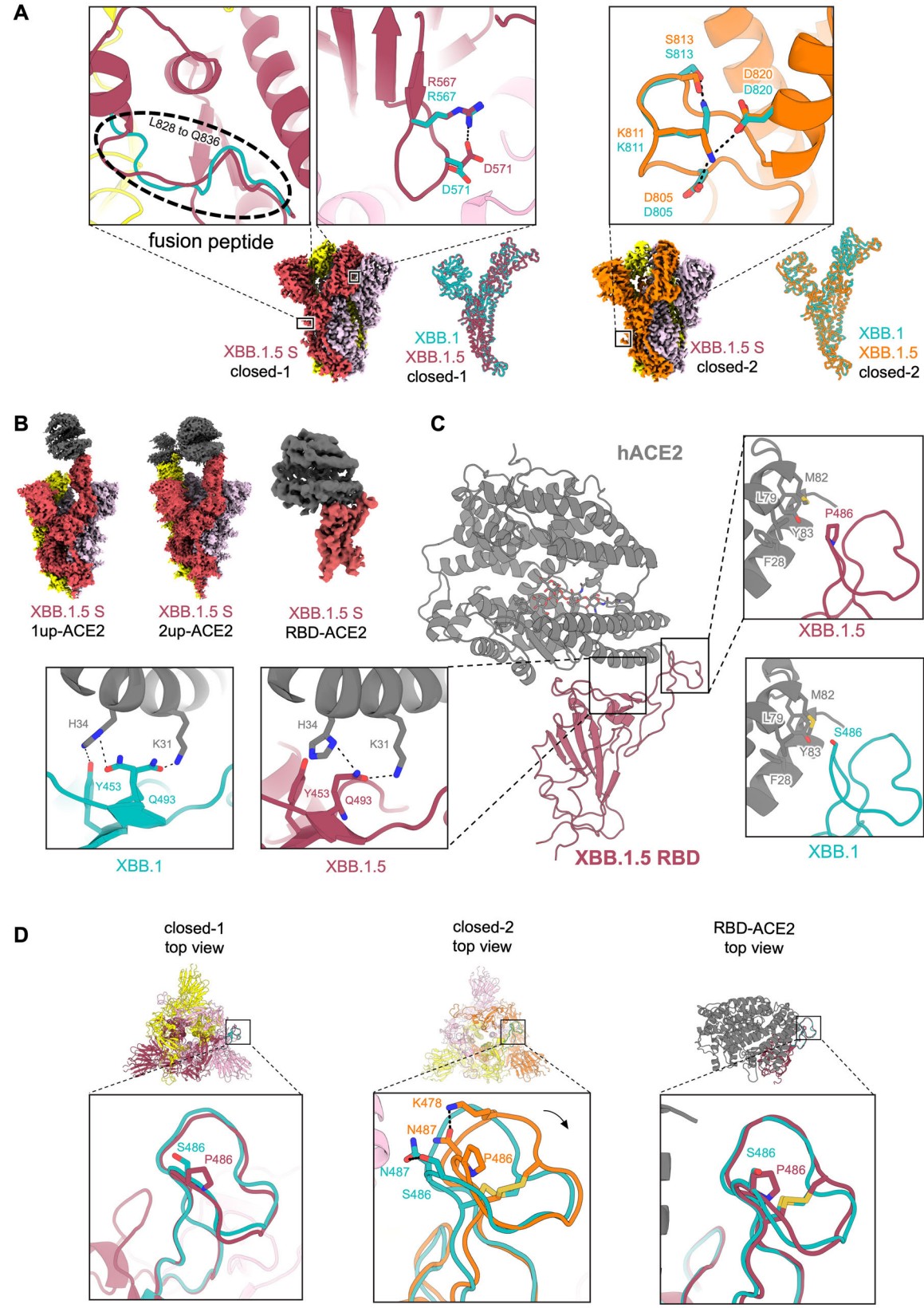

the bottom channel of the XBB1.5-infected airway-on-a-chip was comparable to that of the XBB.1-infected airway-on-a-chip (Fig. 4G, right). However, the barrier-disrupting capacity of XBB.1.5 was significantly lower than that of Delta. Together with the findings of the S-based fusion assay (Figs. 4A, B, and Supplementary Fig. 3A), these results suggest that the fusogenicity of XBB.1.5 is comparable to that of XBB.1.

## Virological characteristics of XBB.1.5 in vivo

To investigate the virological features of XBB.1.5 in vivo, clinical isolates of Delta, XBB.1, and XBB.1.5 (10,000 TCID$_{50}$) were intranasally inoculated into hamsters under anesthesia. Consistent with our previous studies, Delta infection resulted in weight loss (Fig. 5A, left). The body weights of XBB.1- and XBB.1.5-infected hamsters did not increase

**Fig. 3 | Interaction between XBB.1.5 S and ACE2. A** Cryo-EM maps of XBB.1.5 S protein trimer closed-1 state (**left**) and closed-2 state (**right**). Each protomer is colored raspberry, yellow, and light pink (closed-1) or orange, yellow, and light pink (closed-2). In the close-up views, structures of the XBB.1.5 S protein trimer closed-1 state and closed-2 state are shown in the ribbon and stick model, and the corresponding residues in the XBB.1 S closed-1 and closed-2 structures (PDB: 8IOS and 8IOT, respectively) are also shown in cyan. On the right side of the closed-1 and closed-2 cryo-EM maps, the superposition of the main chain structures of the XBB.1 and XBB.1.5 protomers are shown. Dashed lines represent hydrogen bonds. **B** Cryo-EM maps of the XBB.1.5 S protein (same colors as **A**) bound to human ACE2 (gray) in the one-up state (**left**), two-up state (**middle**), or RBD−ACE2 interface (**right**). (**C**) Structure of the RBD−ACE2 complex (same colors as **B**). In the close-up views, residues involved in the corresponding interaction of the XBB.1.5 RBD−ACE2 complex structure, which is different from the XBB.1 RBD−ACE2 complex structure (PDB, 8IOV; RBD, cyan; ACE2, gray), are shown. The N103-linked glycan of ACE2 is represented by stick models. (**D**) Structures of the XBB.1.5 S protein trimer closed-1 state (**left**) and closed-2 state (**middle**), as well as the monomer RBD−ACE2 (**right**). In the close-up views, the RBD Y473−P491 loops (same colors as **C** and **D**) are shown, and the corresponding residues in the XBB.1 S are also shown as cyan sticks.

compared with the negative control (Fig. 5A, left). The hamsters infected with XBB.1 exhibited statistically more reduction of weight compared with those infected with XBB.1.5. We then analyzed the pulmonary function of infected hamsters as reflected by two parameters, enhanced pause (Penh) and the ratio of time to peak expiratory flow relative to the total expiratory time (Rpef). Of the four groups, Delta infection resulted in significant differences in these two respiratory parameters compared with XBB.1.5 (Fig. 5A, middle and right), suggesting that Delta is more pathogenic than XBB.1.5. In contrast, the Penh and Rpef values of both XBB.1- and XBB.1.5-infected hamsters were slightly deteriorated compared with those of uninfected hamsters, although this was not statistically significant (Fig. 5A, middle and right).

To evaluate the viral spread in infected hamsters, we routinely measured the viral RNA load in the oral swab. At 2 days post-infection (d.p.i), the viral RNA loads of hamsters infected with Delta and XBB.1.5 were significantly higher than those infected with XBB.1, but a significantly lower RNA load was detected from XBB.1.5-infected hamsters at 5 d.p.i. (Fig. 5B, **left**). To assess the differences in viral spreading between XBB.1 variants in respiratory tissues, we collected the lungs of infected hamsters at 2 and 5 d.p.i., and the collected tissues were separated into the hilum and periphery regions. The viral RNA loads in both the lung hilum and periphery of Delta-infected hamsters were significantly higher than those of the two XBB.1 variants. The viral RNA loads in both lung regions of XBB.1.5-infected hamsters were slightly higher than those of XBB.1-infected hamsters (Fig. 5B, middle and right) and the infectious titers in the lungs exhibited the similar tendency of viral RNA loads (Supplementary Fig. 4A), suggesting that the viral dissemination of XBB.1.5 in the lungs is slightly higher than that of the ancestral XBB.1, and still significantly lower than that of Delta. To further investigate the viral spread in the respiratory tissues of infected hamsters, we analyzed the formalin-fixed right lungs of infected hamsters at 2 and 5 d.p.i. by carefully identifying the four lobules (and main bronchus), lobar bronchi sectioning each lobe along with the bronchial branches and performing immunohistochemical (IHC) analysis targeting the viral nucleocapsid (N) protein. Consistent with our previous studies[1,3,4,21–24], in the alveolar space around the bronchi/bronchioles at 2 d.p.i., N-positive cells were detected in Delta-infected hamsters (Fig. 5C). The percentage of N-positive cells in the lungs of XBB.1- and XBB.1.5-infected hamsters were relatively low and comparable (Figs. 5C, 2 d.p.i.), but differed in the efficiency of spreading to the alveoli. At 2 d.p.i., the XBB.1.5 N-positive cells were sporadic on the bronchial epithelium and had begun to spread into the alveolar space. However, those of XBB.1 remained strongly and continuously in the bronchial epithelium (Fig. 5C), and consistent with this, bronchitis/bronchiolitis remained high (Figs. 5D and 5E). At 5 d.p.i., N-positive cells were detected in the peripheral alveolar space in Delta-infected hamsters, while the N-positive areas of XBB.1- and XBB.1.5-infected hamsters were sporadic and slightly detectable (Figs. 5C, 5 d.p.i.).

## Intrinsic pathogenicity of XBB.1.5
To investigate the intrinsic pathogenicity of XBB.1.5, histopathological scoring was performed based on the criteria described in our previous studies[22]. Consistent with our previous studies[2,22,23], all five histological parameters as well as the total score of the Delta-infected hamsters were significantly greater than those of the XBB.1- and XBB.1.5-infected hamsters at 5 d.p.i. (Fig. 5E). Compared with the histopathological scores of XBB subvariants, the total histopathological score of XBB.1 was higher than that of XBB.1.5 at 2 d.p.i. and eventually become comparable between XBB.1.5 and XBB.1 at 5 d.p.i.. The inflammation area in the lungs of XBB.1.5-infected hamsters was larger than that of XBB.1 (Fig. 5E and Supplementary Fig. 4B−C). However, bronchitis was more severe in XBB.1 than in XBB.1.5 at both 2 d.p.i. and 5 d.p.i. (Fig. 5E). Consisting with this observation, the dynamics of Type II pneumocytes were opposite between these two XBBs. These results support the high bronchial affinity of XBB.1, as we previously reported, and suggest that XBB.1.5 diminishes its properties.

## Effects of the ORF8 KO phenotype in the XBB.1.5 variant
The G8 nonsense mutation in ORF8 of XBB.1.5 results in an inability to synthesize full-length ORF8 protein. The ORF8 protein has been reported to suppress the expression of major histocompatibility complex class I (MHC-I)[25]. Therefore, unlike previous variants, the XBB.1.5 variant may not be capable of sufficient suppression of MHC-I expression. To investigate this possibility, human induced pluripotent stem cell (iPSC)-derived lung organoids were infected with XBB.1, XBB.1.5, or Delta variants (Fig. 6A and Supplementary Fig. 5). As expected, the expression levels of human MHC-I (Human leukocyte antigen class I, HLA-I) were significantly reduced in lung organoids infected with the XBB.1 and Delta variants. The reduction in lung organoids infected with the XBB.1.5 variant was mild (Fig. 6A), and the HLA-I expression levels were similar between XBB.1 and Delta (Fig. 6B). Collectively, these results suggest that the XBB.1.5 variant weakens suppression of HLA-I expression because of the G8 nonsense mutation in ORF8.

## Intrinsic pathogenicity of recombinant XBB.1 bearing a mutation in S and ORF8
To further investigate the effect of the ORF8:G8stop and S:S486P mutations on viral pathogenicity, we generated four mutant viruses—two bearing either ORF8:G8stop or S:S486P on XBB.1 (rXBB.1/ORF8:G8stop and rXBB.1/S:S486P), and recombinant XBB.1 (rXBB.1) and XBB.1.5 (rXBB.1.5)—by the CPER method[26]. Then, the four recombinant viruses were individually inoculated into hamsters as in the above experiment. All hamsters infected with recombinant viruses exhibited a reduction of body weight (Fig. 7A). Although the weight changes were similar, infection with the recombinant virus rXBB.1/S:S486P was significantly different from rXBB.1.5 and comparable to rXBB.1. In contrast, the dynamics of the recombinant virus rXBB.1/ORF8:G8stop were intermediate, between the dynamics of rXBB.1 and rXBB.1.5. The viral RNA loads in oral swabs of infected hamsters were generally similar (Fig. 7B). The percentage of N-positive cells in the lungs of hamsters infected with the four types of virus were generally low and comparable to each other (Fig. 7C and D). As for inflammation in the lungs infected with the recombinant viruses, the recombinant virus bearing rXBB.1/S:S486P evoked strong bronchitis, similar to

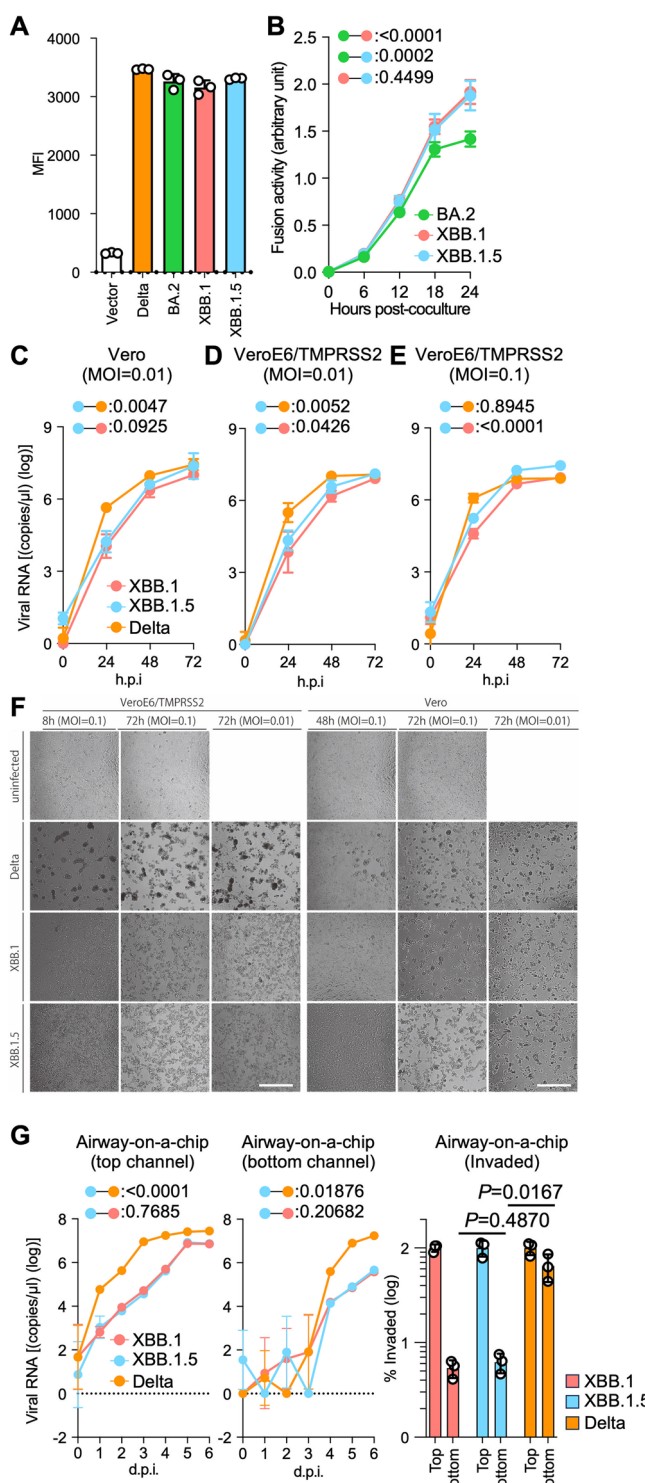

**Fig. 4 | Virological characteristics of XBB.1.5 in vitro.** SARS-CoV-2 S protein-mediated membrane fusion assay in Calu-3/DSP$_{1-7}$ cells. Surface S protein expression level in transfected HEK293 cells (**A**). Fusion activity (arbitrary units) of the BA.2, XBB.1, and XBB.1.5 S proteins are shown (**B**). Growth kinetics of XBB.1.5. Clinical isolates of Delta, XBB.1, and XBB.1.5 were inoculated into Vero cells (**C**) and VeroE6/TMPRSS2 cells (**D** and **E**). The copy numbers of viral RNA in the culture supernatant (**C–E**) were routinely quantified by RT-qPCR. (**F**) The cells were observed under microscopy to judge the CPE appearance. Scale bar: 500 μm. (**G**) Clinical isolates of XBB.1, XBB.1.5, and Delta were inoculated into an airway-on-a-chip system. The copy numbers of viral RNA in the top (**left**) and bottom (**middle**) channels of an airway-on-a-chip were routinely quantified by RT-qPCR. On the **right**, the percentage of viral RNA load in the bottom channel per top channel at 6 d.p.i. (i.e., % invaded virus from the top channel to the bottom channel) is shown. Assays were performed in triplicate (**A, G**) or quadruplicate (**B–F**). The presented data are expressed as the average ± SD (**A, B**) or SEM (**C–G**). Statistically significant differences versus XBB.1 across timepoints were determined by multiple regression (**B-E** and **G left**) or a two-sided Student's *t* test (**G right**). The FWERs calculated using the Holm method are indicated in the figures. NA, not applicable.

XBB.1.5 possesses an effective reproduction number (R$_e$) 1.2 times higher than that of XBB.1 and exhibits significantly higher affinity to human ACE2 than XBB.1[6]. In addition, Halfmann et al. recently reported that XBB.1.5 exhibited higher transmission ability than the ancestral BA.2 in a hamster model[27]. XBB.1.5 has two additional mutations—the S486P mutation in S and the G8 nonsense mutation in ORF8—compared with the original XBB.1 (Fig. 1). Our recent report showed that the S486P mutation in S is a determinant of enhanced affinity to human ACE2[6]. However, the effects of the G8 nonsense mutation in ORF8 on viral properties have not been elucidated. In this study, we characterized the virological features of XBB.1.5, particularly focusing on the impacts of these two mutations. Of note, the XBB.1 strain used in the present study (the TY41-795 strain) harbors serine at 486 in S (S486), and thus we investigated the differences in the properties for either S or P at this locus in S.

The level of antiviral humoral immunity induced by prior vaccination was assessed (Fig. 2). Currently, three (monovalent, BA.1 bivalent, and BA.5 bivalent) types of vaccinations are implemented in humans[28]. Because immunoprophylaxis against SARS-CoV-2 is critical for further implementation for controlling disease, several groups have reported immune evasion of the neutralizing antibody response by XBB.1.5[6,8–15], suggesting comparable properties of immune evasion between XBB.1 and XBB.1.5. Here, we also showed that, regardless of the type of bivalent mRNA vaccines, not only XBB.1 but also XBB.1.5 are robustly resistant to vaccine sera. Sera obtained from monovalent vaccinees could protect against an early-pandemic B.1.1 infection but not against either XBB.1 or XBB.1.5. XBB.1.5 significantly evades four-dose monovalent vaccine sera (28.6-fold, *P* < 0.0001), BA.1 bivalent vaccine sera (34.0-fold, *P* < 0.0001), and BA.5 bivalent vaccine sera (16.8-fold, *P* < 0.0001), more effectively than B.1.1. Collectively, these data indicate that no further immune escape resulted from the evolution of XBB.1 into XBB.1.5.

By cryo-EM analysis, a structural comparison of the S proteins of XBB.1 and XBB.1.5 in closed-1 revealed a difference in the loop structure consisting of L828 to Q836 within the fusion peptide (residues 816–855) (Fig. 3A). In contrast, in closed-2, the cryo-EM maps around the fusion peptide in both the XBB.1 and XBB.1.5 S proteins were highly disordered. This structural diversity in the fusion peptide suggests that this region is highly mobile. Despite a structural difference in the fusion peptide portion, which is essential for membrane fusion, the overall structures of XBB.1 and XBB.1.5 were highly similar. Taken together, these data correspond to our data on viral infection; the membrane fusion ability, antibody evasion ability, and in vitro replication properties are comparable between XBB.1. and XBB.1.5. Indeed, the fusion ability was similar between XBB.1 and XBB.1.5 and greater than BA.2 (Fig. 4B). In cell culture, XBB.1.5 and XBB.1 exhibited

rXBB.1, whereas that bearing rXBB.1/ORF8:G8stop caused mild bronchitis, resembling XBB.1.5 (Fig. 7E and F). Taken together, these data suggest that both mutations, S:S486P and ORF8:G8stop alter viral function and are involved in viral pathogenicity in hamsters.

## Discussion

Although people are naturally infected with SARS-CoV-2 and/or sufficiently vaccinated, SARS-CoV-2 variants are emerging and spreading among humans. In the last two years, Omicron subvariants have shown a variety of evolutionary traces[3,4], and XBB.1.5 became a dominant variant in the USA in early 2023. Our previous report showed that

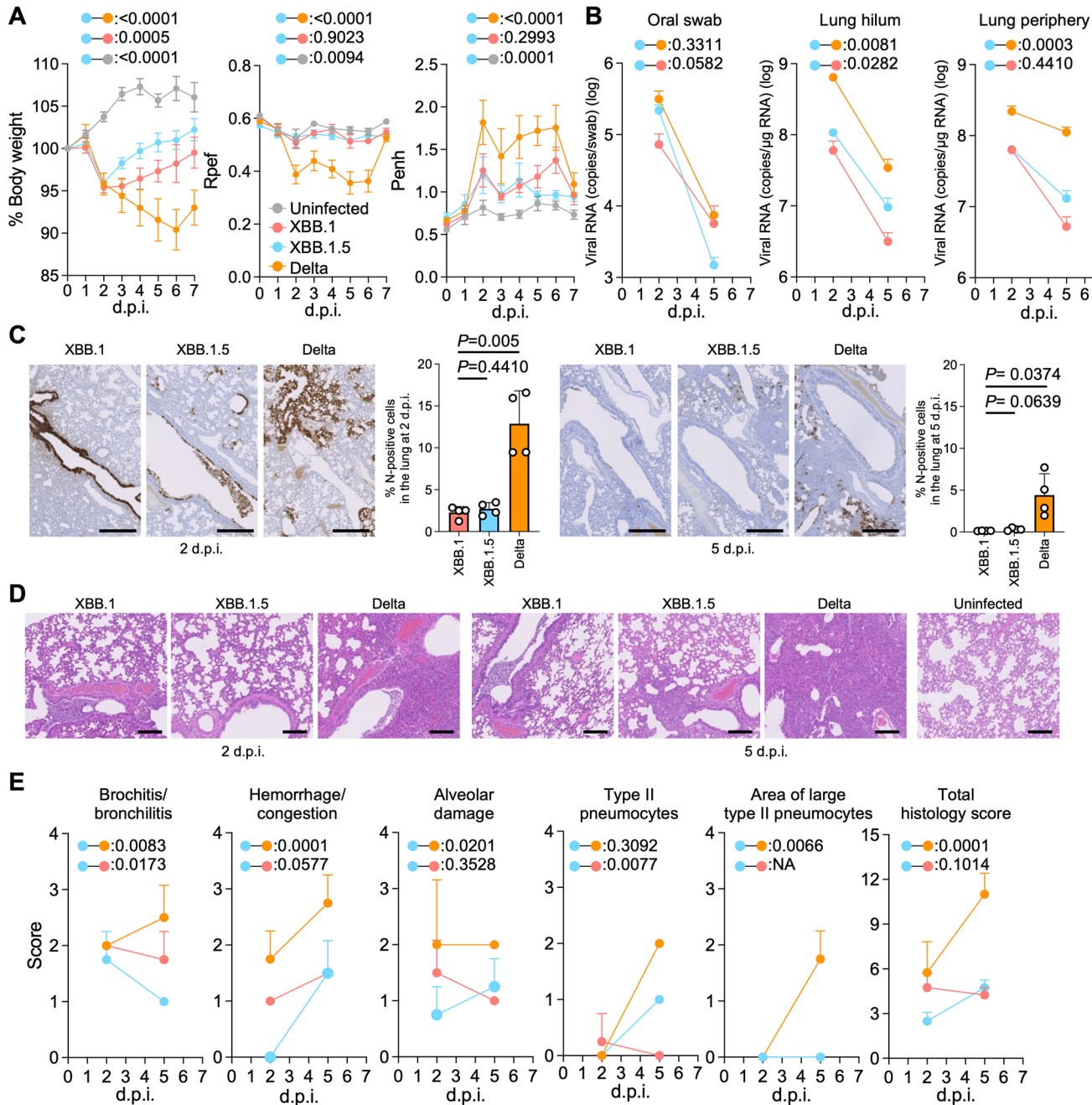

**Fig. 5 | Virological characteristics of XBB.1.5 in vivo.** Syrian hamsters were intranasally inoculated with XBB.1, XBB.1.5, and Delta. Six hamsters of the same age were intranasally inoculated with saline (uninfected). Six hamsters per group were used to routinely measure the respective parameters (**A**). Four hamsters per group were euthanized at 2 and 5 d.p.i. and used for virological and pathological analysis (**B–E**). **A** Body weight, Penh, and Rpef values of infected hamsters (*n* = 6 per infection group). **B Left** Viral RNA loads in the oral swab (*n* = 6 per infection group). (**Middle** and **right**) Viral RNA loads in the lung hilum (**middle**) and lung periphery (**right**) of infected hamsters (*n* = 4 per infection group). **C** IHC of the viral N protein in the lungs of infected hamsters at 2 d.p.i. (**left**) and 5 d.p.i. (**right**). Representative figures (N-positive cells are shown in brown) and the percentage of N-positive cells in whole lung lobes (*n* = 4 per infection group) are shown. **D** H&E staining of the lungs of infected hamsters. Representative figures are shown in (**D**). Uninfected lung alveolar spaces are also shown. The raw data are shown in Supplementary Fig. 4B and Supplementary Fig. 4C. **E** Histopathological scoring of lung lesions (*n* = 4 per infection group). Representative pathological features are reported in our previous studies[1,3,4,21–24]. **A–C, E** Data are presented as the average ± SEM. (**C**) Each dot indicates the result of an individual hamster. **A, B, E** Statistically significant differences between XBB.1.5 and other variants across timepoints were determined by multiple regression. **B, E** The 0 d.p.i. data were excluded from the analyses. The FWERs calculated using the Holm method are indicated in the figures. **C** The statistically significant differences between XBB.1.5 and other variants were determined by a two-sided Mann–Whitney *U* test. **C** and **D** Each panel shows a representative result from an individual infected hamster. Scale bars, (**C**) 500 µm, (**D**) 200 µm.

comparable growth rates, but in VeroE6/TMPRSS2 cells, XBB.1.5 showed a slightly higher growth rate (Fig. 4C–E and Supplementary Fig. 3B–D). Regarding morphological changes upon infection with SARS-CoV-2, Delta infection resulted in large multinuclear syncytia and Omicron subvariants displayed small syncytia, but XBB.1.5

formed an intermediate number of syncytia (Fig. 4F), supporting data showing a stronger affinity of XBB.1.5 S protein.

Because XBB.1.5 has evolved and exhibits dominance in humans, airway-on-a-chip was employed to evaluate whether XBB.1.5 has an enhanced ability to disrupt the respiratory epithelial and endothelial

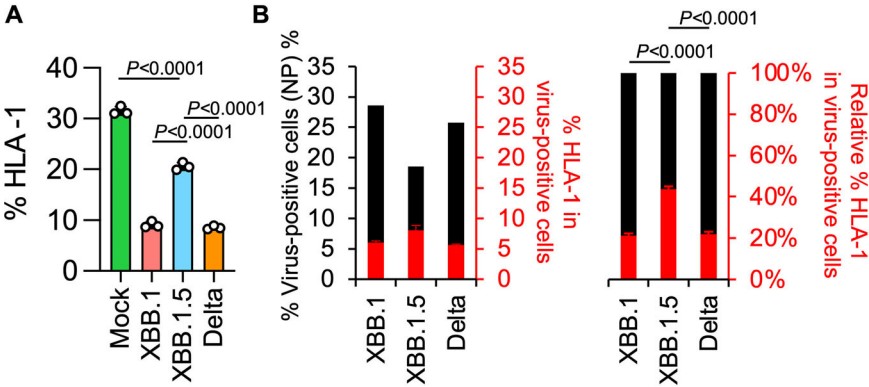

**Fig. 6 | Effects of ORF8 KO on HLA-I expression in human lung organoids.**
**A** Clinical isolates of XBB.1, XBB.1.5, and Delta were inoculated into human iPSC-derived lung organoids. The percentages of HLA-I-positive cells in the lung organoids are shown. **B** The percentage of HLA-I-positive cells in viral N protein-positive lung organoids is shown. Assays were performed in triplicate. The presented data are expressed as the average ± SD. The statistically significant differences between XBB.1.5 and other variants were determined by a two-sided Student's *t* test.

barriers. Interestingly, different yet non-significant characteristics were observed between XBB.1.5 and XBB.1 (Fig. 4G and Supplementary Fig. 3G). These data indicate that, although XBB.1.5 has acquired an additional mutation in S, S486P, its replication capability in respiratory cells is similar to the ancestral XBB.1.

Viral intrinsic pathogenicity was assessed in the hamster model, which has been established and employed in our series of studies[1–5,21,22,24,29,30]. XBB.1.5 exhibited a smaller reduction of body weight compared to XBB.1 and Delta (Fig. 5A). XBB.1.5 infection and XBB.1 infection were comparable in terms of viral load in the respiratory organs to XBB.1.5 (Fig. 5B and Supplementary Fig. 4A). On histopathological analysis, the severity of bronchitis/bronchiolitis was diminished upon infection with XBB.1.5 (Figs. 5D, E and Supplementary Fig. 4B, C). Although XBB1.5 might have acquisition of enhanced transmissibility, these data indicate that the acquisition of the S:S486P and ORF8 stop mutations are involved in the pathogenicity of XBB.1.5. In a previous study, Kohyama et al. showed that ORF8 functions as a viral cytokine and that functional loss of ORF8 resulted in impairment of inflammation[31]. As shown in Fig. 6A, Delta and the ancestral XBB.1, but not XBB.1.5, suppressed the expression of MHC-I in human iPSC-derived lung organoids. In vivo analysis using the dysfunctional recombinant ORF8 in the XBB.1 backbone, as shown in Fig. 7A, revealed that the reduction of body weight was impaired and comparable to XBB.1.5. Histopathological analyses also indicated suppression of inflammation (Fig. 7E and F). In particular, ORF8 dysfunction resulted in remission of bronchitis/bronchiolitis, indicating that severe bronchiolitis/bronchiolitis upon XBB.1 infection leads to the reduction of body weight. Taken together, this indicates that functional loss of ORF8 is involved in the attenuation profile of in vivo pathogenesis of XBB.1.5. In our previous study and those by other groups[6,9,27], XBB.1.5 may have acquired greater transmission ability compared with the ancestral variant. We have two reasons to conclude that XBB.1.5 might have acquisition of enhanced transmissibility. First, the Spike-hACE2 complex of XBB.1 and XBB.1.5 showed that binding affinity is enhanced by S486P mutation in XBB.1.5 as discussed above. Secondly, in hamsters, XBB.1.5 possesses slightly higher ability of viral shedding and dissemination compared with the ancestral XBB.1. However, because XBB.1.5 acquired the mutation to disrupt the function of ORF8 and did not suppress MHC-1 efficiently, the virus shedding impaired at the later time point (Fig. 5B). Moreover, our series of studies on continued in-depth viral genomic surveillance and real-time evaluation of the risk of newly emerging SARS-CoV-2 variants[1–4,21–24], together with the present study, suggest that Omicron subvariants have evolved to 'trade off' two capabilities: increased ACE2 binding, and reduced immunopathogenesis.

In summary, our comprehensive characterization of the newly emerged XBB.1.5 strain suggests that the Omicron subvariant is evolving enhanced viral spreading with diminished viral characteristics, to aid survival in humans. Our findings both improve our understanding of SARS-CoV-2 ecology and will aid implementation of the correct strategy for controlling COVID-19.

## Methods

### Ethics statement

All experiments with hamsters were performed in accordance with the Science Council of Japan's Guidelines for the Proper Conduct of Animal Experiments. The protocols were approved by the Institutional Animal Care and Use Committee of National University Corporation Hokkaido University (approval IDs: 20-0123 and 20-0060).

All human subjects in this study were recruited at Interpark Kuramochi Clinic with written informed consent, which allows publication of more than three indirect identifiers. All protocols involving samples from human subjects recruited at Interpark Kuramochi Clinic were reviewed and approved by the Institutional Review Board of Interpark Kuramochi Clinic (approval ID: G2021-004). All protocols for the use of human specimens were reviewed and approved by the Institutional Review Boards of The Institute of Medical Science, The University of Tokyo (approval IDs: 2021-1-0416 and 2021-18-0617), Kumamoto University (approval ID: 2066), and University of Miyazaki (approval ID: O-1021).

### Human serum collection

Fourth-dose vaccine sera from individuals who had been vaccinated with monovalent vaccine (15 donors; average age: 42 years, range: 30–56 years, 40% male) (Fig. 2A), BA.1 bivalent vaccine sera (20 donors; average age: 55 years, range: 30–80 years, 35% male) (Fig. 2B), and BA.5 bivalent vaccine sera (21 donors; average age: 51 years, range: 18–86 years, 48% male) (Fig. 2C) were collected. The SARS-CoV-2 variants were identified as previously described[3,21]. Sera were inactivated at 56 °C for 30 minutes and stored at −80 °C until use. The details of the convalescent sera are summarized in Supplementary data 2.

### Cell culture

HEK293T cells (a human embryonic kidney cell line; ATCC, CRL-3216), HEK293 cells (a human embryonic kidney cell line; ATCC, CRL-1573) and HOS-ACE2/TMPRSS2 cells (HOS cells stably expressing human ACE2 and TMPRSS2)[32,33] were maintained in DMEM (high glucose) (Sigma-Aldrich, Cat# 6429-500 M; Nacalai Tesque Cat# 08458-16) containing 10% fetal bovine serum (FBS, Sigma-Aldrich Cat# 172012-500 ML) and 1% penicillin–streptomycin (PS) (Sigma-Aldrich, Cat# P4333-100ML). Vero cells (an African green monkey (*Chlorocebus*

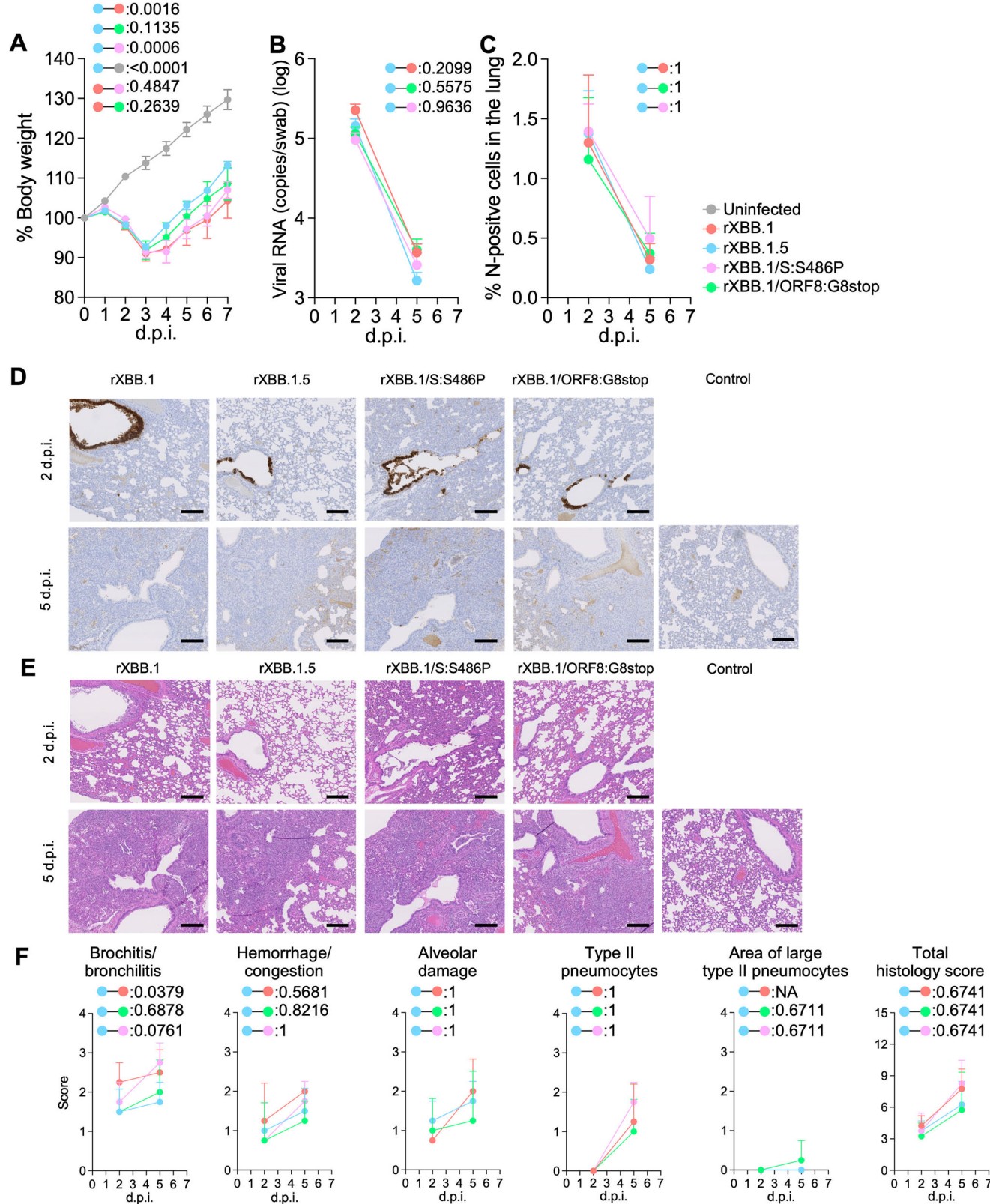

sabaeus) kidney cell line; JCRB Cell Bank, JCRB0111) were maintained in Eagle's minimum essential medium (EMEM) (Sigma-Aldrich, Cat# M4655-500ML) containing 10% FBS and 1% PS. VeroE6/TMPRSS2 cells (VeroE6 cells stably expressing human TMPRSS2; JCRB Cell Bank, JCRB1819)[34] were maintained in DMEM (low glucose) (FUJIFILM WAKO, Cat# 041-29775) containing 10% FBS, G418 (1 mg/ml; Nacalai Tesque, Cat# G8168-10ML), and 1% PS. Calu-3 cells (ATCC, HTB-55) were

maintained in Eagle's minimum essential medium (EMEM) (Sigma-Aldrich, Cat# M4655-500ML) containing 10% FBS and 1% PS. Calu-3/DSP$_{1-7}$ cells (Calu-3 cells stably expressing DSP$_{1-7}$)[35] were maintained in EMEM (FUJIFILM WAKO, Cat# 056-08385) containing 20% FBS and 1% PS. HEK293S GnTI(−) cells (HEK293S cells lacking N-acetylglucosaminyltransferase)[36] were maintained in DMEM (high glucose) containing 2% FBS.

**Fig. 7 | Virological characteristics of recombinant viruses bearing the single mutations in S and ORF8 in vivo.** Syrian hamsters were intranasally inoculated with the recombinant viruses rXBB.1, rXBB.1/S:S486P, rXBB.1/ORF8:G8stop, and rXBB.1.5. Six hamsters of the same age were intranasally inoculated with saline (uninfected). Six hamsters per group were used to routinely measure the body weight (**A**). Four hamsters per group were euthanized at 2 and 5 d.p.i. and used for virological and pathological analysis (**B–E**). **A** Body weight of infected hamsters (*n* = 6 per infection group). **B** Viral RNA loads in the oral swab (*n* = 4 per infection group). **C** Percentage of the viral N-positive cells in the lungs at 2 and 5 d.p.i. of infected hamsters. Representative figures (N-positive cells are shown in brown) are shown in (**D**). Uninfected lung alveolar space and bronchioles are also shown. **E** H&E staining of the lungs of infected hamsters. Representative images showing the same area with the viral N staining as in (**D**). **F** Histopathological scoring of lung lesions (*n* = 4 per infection group). Representative pathological features are reported in our previous studies[1–4,21–24]. (**A–C**, and **F**) Data are presented as the average ± SEM.

## Viral genome sequencing

The sequences of all four recombinant viruses generated in this study were confirmed by sequencing with a SeqStudio Genetic Analyzer (Thermo Fisher Scientific) and an outsourced service (Fasmac).

## Phylogenetic tree and ancestral genomic sequence reconstruction

We obtained surveillance data of 14,617,387 SARS-CoV-2 isolates from the GISAID database on January 24, 2023 (https://www.gisaid.org)[37]. The PANGO lineage of each SARS-CoV-2 isolate was also assigned using NextClade v2.14.0[38] in parallel. We excluded the data of any SARS-CoV-2 isolate that i) lacked GISAID and NextClade PANGO lineage information; ii) was isolated from non-human hosts; iii) was sampled from the original passage; and iv) whose genomic sequence was no longer than 28,000 base pairs and contained ≥2% of unknown (N) nucleotides. In total, the filtered data contain the data of 29,608 SARS-CoV-2 isolates in XBB lineage. We used the GISAID PANGO lineage classification in downstream analyses.

We performed random sampling to retrieve up to 10 genomic sequences from the combination of each XBB sublineage and each country, resulting in a total of 2,350 genomic sequences of SARS-CoV-2. The sampled genomic sequences were then aligned to the genomic sequence of Wuhan-Hu-1 SARS-CoV-2 isolate (NC_045512.2) with reference-guide multiple pairwise alignment strategy using ViralMSA v1.1.24[39]. Gaps in the alignment were removed automatically using TrimAl v1.4.rev22 with -gappyout mode[40]. A preliminary maximum likelihood-based phylogenetic tree of representative XBB sublineages was reconstructed from the alignment using IQ-TREE v2.2.0[41]. The best-fit nucleotide substitution model was selected automatically using ModelFinder implemented in the IQ-TREE suite[42]. Branch support was assessed using and Shimodaira-Hasegawa-like approximate likelihood ratio test and ultrafast bootstrap approximation[43] with 1,000 replicates. We subsequently removed genomic sequences causing branch length outliers in the preliminary tree determined by Rosner test implemented in the EnvStats R package v2.7.0[44] using R v4.2.2[45]. The final tree was then reconstructed (EPI SET ID: EPI_SET_231124cy) using the methods described earlier. The tree was visualized using ggtree R package v3.6.2[46]. The XBB sublineage was used as an outgroup for tree rooting. The ancestral genomic sequence was reconstructed from the genomic sequences without gap trimming using empirical Bayesian method implemented in the IQ-TREE suite, which consider both branch length and nucleotide substitution model[41]. The best-fit model used in the ancestral genomic sequence reconstruction was the same model used in the phylogenetic tree reconstruction.

## Sensitivity analysis on the effect of genomic sequence down-sampling

We performed the sensitivity analysis to assess the effect of genomic sequence down-sampling on reliability of phylogenetic analyses. Genomic sequences of SARS-CoV-2 were randomly sampled ten times to generate ten different datasets (Supplementary data 1). We then performed reconstructions of phylogenetic tree and ancestral genomic sequence as described earlier. Topology of the phylogenetic trees and the order of mutation occurrences were subsequently compared.

## Plasmid construction

Plasmids expressing the codon-optimized SARS-CoV-2 S proteins of B.1.1 (the parental D614G-bearing variant), BA.2, BA.5, XBB.1, and XBB.1.5 were prepared in our previous studies[1,3,4,6]. The resulting PCR fragments were digested with KpnI (New England Biolabs, Cat# R0142S) and NotI (New England Biolabs, Cat# R1089S) and inserted into the corresponding site of the pCAGGS vector[47]. To generate recombinant SARS-CoV-2, the nine pmW118 plasmid vectors were subjected to amplification of the cDNA fragments (F1–F9-10) of SARS-CoV-2 XBB.1 and XBB.1.5 by PrimeSTAR GXL DNA polymerase (Takara) with the primer sets[26]. To generate mutant plasmids bearing mutations (S:S486P and ORF8:G8stop), inverse PCR using the primer listed in Supplementary data 3 was performed. Then, the resultant PCR product was treated with *Dpn*I and then subjected to self-ligation and transformation for obtaining the desired plasmids. Nucleotide sequences were confirmed by the Sanger method as described above.

## Neutralization assay

Pseudoviruses were prepared as previously described[1,3,4,22,24,29,33,48,49]. Briefly, lentivirus (HIV-1)-based, luciferase-expressing reporter viruses were pseudotyped with SARS-CoV-2 S proteins. HEK293T cells (1,000,000 cells) were cotransfected with 1 μg psPAX2-IN/HiBiT[32], 1 μg pWPI-Luc2[32], and 500 ng plasmids expressing parental S or its derivatives using PEI Max (Polysciences, Cat# 24765-1) or TransIT-293 (Takara, Cat# MIR2700) with Opti-MEM (Thermo Fisher Scientific, Cat# 11058021) according to the manufacturer's protocol. Two days post-transfection, the culture supernatants were harvested and centrifuged. The pseudoviruses were stored at −80 °C until use.

The neutralization assay (Fig. 2) was prepared as previously described[1,3,4,22,24,33,49–51]. Briefly, the SARS-CoV-2 S pseudoviruses (counting ~20,000 relative light units) were incubated with serially diluted (120-fold to 87,480-fold dilution at the final concentration) heat-inactivated sera at 37 °C for 1 hour. Pseudoviruses without sera were included as controls. Then, a 40 μl mixture of pseudovirus and serum/antibody was added to HOS-ACE2/TMPRSS2 cells (10,000 cells/50 μl) in a 96-well white plate. At 2 d.p.i., the infected cells were lysed with a One-Glo luciferase assay system (Promega, Cat# E6130), a Bright-Glo luciferase assay system (Promega, Cat# E2650), or a britelite plus Reporter Gene Assay System (PerkinElmer, Cat# 6066769), and the luminescent signal was measured using a GloMax explorer multi-mode microplate reader 3500 (Promega) or CentroXS3 LB960 (Berthold Technologies). The assay of each serum sample was performed in triplicate, and the 50% neutralization titer (NT$_{50}$) was calculated using Prism 9 software v9.1.1 (GraphPad Software).

## SARS-CoV-2 preparation and titration

The working virus stocks of SARS-CoV-2 were prepared and titrated as previously described[1,24,29]. In this study, clinical isolates of Delta (B.1.617.2, strain TKYTK1734; GISAID ID: EPI_ISL_2378732)[22], XBB.1 (strain TY41-795; GISAID ID: EPI_ISL_15669344)[3] and XBB.1.5 (strain TKYmbr30523/2022; GISAID ID: EPI_ISL_16697941) were used. In brief, 20 μl of the seed virus was inoculated into VeroE6/TMPRSS2 cells (5,000,000 cells in a T-75 flask). At 1 h.p.i., the culture medium was replaced with DMEM (low glucose) (FUJIFILM WAKO, Cat# 041-29775) containing 2% FBS and 1% PS. At 3 d.p.i., the culture medium was

harvested and centrifuged, and the supernatants were collected as the working virus stock.

Recombinant viruses were generated by a circular polymerase extension reaction (CPER) as previously described with modification[26]. The nine fragments of SARS-CoV-2 and the UTR linker for SARS-CoV-2 were prepared by PCR using PrimeSTAR GXL DNA polymerase (Takara). After gel purification of the fragments, CPER was conducted by a thermal cycler. The CPER products (25 μl) were transfected into VeroE6/TMPRSS2 cells with TransIT-X2 (Takara, Cat# MIR6003) according to the manufacturer's protocol. When CPE was observed, supernatants were harvested and generated stocks described above. All the viruses were kept at −80 °C until use.

The titer of the prepared working virus was measured as the 50% tissue culture infectious dose ($TCID_{50}$). Briefly, one day before infection, VeroE6/TMPRSS2 cells (10,000 cells) were seeded into a 96-well plate. Serially diluted virus stocks were inoculated into the cells and incubated at 37 °C for four days. The cells were observed under a microscope to judge the CPE appearance. The value of $TCID_{50}$/ml was calculated with the Reed–Muench method[52].

**Plaque assay**

Plaque assay was performed as previously described[29]. Briefly, 1 day before infection, Vero cells or VeroE6/TMPRSS2 cells (100,000 cells per well) were seeded into a 24-well plate and infected with SARS-CoV-2 (0.5, 5, 50, or 500 $TCID_{50}$) at 37 °C for 2 hours. A mounting solution containing 3% FBS and 1.5% carboxymethyl cellulose (Sigma-Aldrich, Cat#C4888-500G) was overlaid, followed by incubation at 37 °C. At 3 d.p.i., the culture medium was removed, and the cells were washed with PBS three times and fixed with 4% paraformaldehyde phosphate buffer solution (Nacalai Tesque, Cat# 09154-85). The fixed cells were washed with tap water, dried, and stained with a staining solution [2% Crystal Violet (Nacalai Tesque, Cat# 09804-52) in water] for 30 minutes. The stained cells were washed with tap water and dried, and the size of the plaques was measured using Adobe Photoshop 2024 v25.0.0 (Adobe).

**Viral ToxGlo assay**

Vero cells or VeroE6/TMPRSS2 cells (18,000 cells per well) were seeded into a 96-well plate. After overnight incubation, cells were infected with SARS-CoV-2 (90, or 900 $TCID_{50}$). At 72 h.p.i., the viral-induced cytopathic effects (CPE) were quantified with Viral ToxGlo Assay (Promega, Cat# G8942) according to the manufacturer's protocol.

**Protein expression and purification for cryo-EM**

The XBB.1.5 S protein ectodomain and human ACE2 were expressed and purified as previously described[53]. Briefly, the expression plasmids, pHLsec, encoding the XBB.1.5 S protein ectodomain with six proline substitutions (F817P, A892P, A899P, A942P, K986P, and V987P)[54] and deletion of the furin cleavage site (i.e., RRAR to GSAG substitution) with a T4-foldon domain or soluble human ACE2 ectodomain, were transfected into HEK293S GnTI(−) cells. Expressed proteins in the cell-culture supernatant were purified using a cOmplete His-Tag Purification Resin (Roche, Cat# 5893682001) affinity column, followed by Superose 6 Increase 10/300 GL size-exclusion chromatography (Cytiva, Cat# 29091596) with calcium- and magnesium-free PBS buffer.

**Cryo-EM sample preparation and data collection**

XBB.1.5 S protein solution was incubated at 37 °C for 1 h prior to cryo-EM grid preparation. Purified ACE2 was incubated with XBB.1.5 S protein at a molar ratio of 1:3.2 (S:ACE2) at 18 °C for 10 min. The samples were then applied to a Quantifoil R2.0/2.0 Cu 300 mesh grid (Quantifoil Micro Tools GmbH). The grid was freshly glow-discharged for 60 sec at 10 mA using a PIB-10 (Vacuum Device). Samples were plunged into liquid ethane using a Vitrobot mark IV (Thermo Fisher

Scientific) with the following settings: temperature, 18 °C; humidity, 100%; blotting time, 5 sec; and blotting force, 5.

Movies were collected at a nominal magnification of 130,000 (0.67 per physical pixel), using a GIF-Biocontinuum energy filter (Gatan) with a 20-eV slit width on a Krios G4 (Thermo Fisher Scientific) operated at 300 kV with a K3 direct electron detector (Gatan). Each micrograph was collected with a total exposure of 1.5 s and a total dose of 56.0 e/$\text{Å}^2$ over 50 frames. A total of 3522 movies for XBB.1.5 S and a total of 3762 (dataset 1) and 3712 (dataset 2) movies for the XBB.1.5 S–ACE2 complexes were collected at a nominal defocus range of 0.7–1.9 μm using EPU software (Thermo Fisher Scientific).

**Cryo-EM image processing**

For the XBB.1.5 S protein trimer alone, movie frames were aligned, dose-weighted through RELION 4.1[55] implementation, and CTF-estimated using CTFFIND4[56]. First, particles were auto-picked based on a Laplacian-of-Gaussian filter, and 2D classification was performed for TOPAZ[57] training. We used 6729 particles in 100 micrographs for training the TOPAZ neural network, and 1,201,208 particles were picked from all micrographs with the trained TOPAZ neural network. Reference-free 2D classification (gradient-driven algorithm, K = 150, T = 2) was performed to remove junk particles. We used 384,747 particles for initial model reconstruction and 3D classification. Two classes of closed states (closed-1 and closed-2) with different RBD orientations were separated in 3D classification. A final map of the closed-1 state was reconstructed by 3D refinement with C3 symmetry following CTF refinement and Bayesian Polishing. The closed-2 state was also processed by 3D refinement with C3 symmetry following CTF refinement and Bayesian Polishing; however, the density of the RBD region was unclear. The particles belonging to the closed-2 state were symmetry-expanded under C3 symmetry operation and transferred to cryoSPARC v4.1.2[58], 3D classification (K = 4, force hard classification, input mode = simple) focused on the RBD without alignment was performed, and the selected classes clearly showed the RBD and a different conformation to the closed-1 state. A final map of the closed-2 state was reconstructed with non-uniform refinement after removing duplicate particles. To support model building, a local refinement focusing on the RBD in the closed-2 state was carried out.

For XBB.1.5 S protein bound to ACE2, movie frames were aligned, dose-weighted, and CTF-estimated using Patch Motion correction and Patch CTF in cryoSPARC v4.1.2. We blob-picked 1,082,622 (dataset 1) and 748,819 (dataset 2) particles, and reference-free 2D classification (K = 150, batch = 200, Iteration = 30) was performed on each dataset separately to remove junk particles. The initial model was reconstructed using particles belonging to dataset 1, and heterogeneous refinement was separately performed for all picked particles in each dataset. To address the flexibility of the RBD–ACE2 interface, a 3D classification (K = 4, force hard classification, input mode = simple) focused on the RBD–ACE2 interface without alignment was performed. A local map of the RBD–ACE2 interface was obtained by local refinement using particles belonging to the class with clearly observed ACE2. Since the down RBD showed multiple conformations, a further 3D classification focused on the down RBD without alignment was performed, and the final maps of the one-up and two-up state global map were reconstructed by non-uniform refinement.

The reported global resolutions are based on the gold-standard Fourier shell correlation curves (FSC = 0.143) criterion. Local resolutions were calculated with cryoSPARC[59]. Data processing workflows are shown in Figure S5. Figures related to data processing and reconstructed maps were prepared with UCSF Chimera (version 1.15)[60] and UCSF Chimera X (version 1.4)[61].

**Cryo-EM model building and analysis**

Structures of the SARS-CoV-2 XBB.1 S protein closed-1 state (PDB:8IOS[3]) and closed-2 state (PDB: 8IOT[3]), as well as the XBB.1

S−ACE2 complex RBD one-up state (PDB: 8IOU[3]) and RBD two-up state, were fitted to the corresponding maps using UCSF Chimera (version 1.16). Iterative rounds of manual fitting in Coot (version 0.9.8.7)[62] and real-space refinement in Phenix (version 1.20.1)[63] were carried out to improve non-ideal rotamers, bond angles, and Ramachandran outliers. The final model was validated with MolProbity[64]. The structure models shown in surface, cartoon, and stick presentation in the figures were prepared with PyMOL (version 2.5.0) (http://pymol.sourceforge.net).

## SARS-CoV-2 S-based fusion assay

A SARS-CoV-2 S-based fusion assay (Figs. 4A, B and Supplementary Fig. 3A) was performed as previously described[1–5,24,29,59]. Briefly, on day 1, effector cells (i.e., S-expressing cells) and target cells (Calu-3/DSP$_{1-7}$ cells) were prepared at a density of 0.6−0.8 × 10⁶ cells in a 6-well plate. On day 2, for the preparation of effector cells, HEK293 cells were cotransfected with the S expression plasmids (400 ng) and pDSP$_{8-11}$ (Ref. 19) (400 ng) using TransIT-LT1 (Takara, Cat# MIR2300). On day 3 (24 hours post-transfection), 16,000 effector cells were detached and reseeded into a 96-well black plate (PerkinElmer, Cat# 6005225), and target cells were reseeded at a density of 1,000,000 cells/2 ml/well in 6-well plates. On day 4 (48 hours post-transfection), target cells were incubated with EnduRen live cell substrate (Promega, Cat# E6481) for 3 hours and then detached, and 32,000 target cells were added to a 96-well plate with effector cells. *Renilla* luciferase activity was measured at the indicated time points using Centro XS3 LB960. For measurement of the surface expression level of the S protein, effector cells were stained with rabbit anti-SARS-CoV-2 S S1/S2 polyclonal antibody (Thermo Fisher Scientific, Cat# PA5-112048, 1:100). Normal rabbit IgG (Southern Biotech, Cat# 0111-01, 1:100) was used as a negative control, and APC-conjugated goat anti-rabbit IgG polyclonal antibody (Jackson ImmunoResearch, Cat# 111-136-144, 1:50) was used as a secondary antibody. The surface expression levels of S proteins (Fig. 4A, Supplementary Fig. 3A) were measured using a FACS Canto II (BD Biosciences) and the data were analyzed using FlowJo software v10.7.1 (FlowJo LLC). Gating strategy for flow cytometry is shown in left panel of Supplementary Fig. 3A. For the calculation of fusion activity, *Renilla* luciferase activity was normalized to the mean fluorescence intensity (MFI) of surface S proteins. The normalized value (i.e., *Renilla* luciferase activity per the surface S MFI) is shown as fusion activity.

## Preparation of human airway and alveolar epithelial cells from human iPSCs

The air−liquid interface culture of airway and alveolar epithelial cells (Fig. 4G) was differentiated from human iPSC-derived lung progenitor cells as previously described[1–4,21,62,65,66]. Briefly, alveolar progenitor cells were induced stepwise from human iPSCs according to a 21-day and four-step protocol[65]. At day 21, alveolar progenitor cells were isolated with the specific surface antigen carboxypeptidase M and seeded onto the upper chamber of a 24-well Cell Culture Insert (Falcon, #353104), followed by 28-day and 7-day differentiation of airway and alveolar epithelial cells, respectively. Alveolar differentiation medium with dexamethasone (Sigma-Aldrich, Cat# D4902), KGF (PeproTech, Cat# 100-19), 8-Br-cAMP (Biolog, Cat# B007), 3-isobutyl 1-methylxanthine (IBMX) (FUJIFILM WAKO, Cat# 095-03413), CHIR99021 (Axon Medchem, Cat# 1386), and SB431542 (Fujifilm Wako, Cat# 198-16543) was used for the induction of alveolar epithelial cells. PneumaCult ALI (STEMCELL Technologies, Cat# ST-05001) with heparin (Nacalai Tesque, Cat# 17513-96), Y-27632 (LC Laboratories, Cat# Y-5301), and hydrocortisone (Sigma-Aldrich, Cat# H0135) was used for induction of airway epithelial cells.

## Airways-on-a-chip

Airways-on-a-chip (Fig. 4G and Supplementary Fig. 3G) were prepared as previously described[2–4,20]. Human lung microvascular endothelial cells (HMVEC-L) were obtained from Lonza (Cat# CC-2527) and cultured with EGM-2-MV medium (Lonza, Cat# CC-3202). For preparation of the airway-on-a-chip, the bottom channel of a polydimethylsiloxane (PDMS) device was first precoated with fibronectin (3 μg/ml, Sigma-Aldrich, Cat# F1141). The microfluidic device was generated according to our previous report[67]. HMVEC-L cells were suspended at 5,000,000 cells/ml in EGM2-MV medium. Then, 10 μl of suspension medium was injected into the fibronectin-coated bottom channel of the PDMS device. The PDMS device was turned upside down and incubated. After 1 hour, the device was turned over, and EGM2-MV medium was added into the bottom channel. After 4 days, airway organoids (AO) were dissociated and seeded into the top channel. AOs were generated according to our previous report[68]. AOs were dissociated into single cells and then suspended at 5,000,000 cells/ml in the AO differentiation medium. Ten microliters of suspension medium were injected into the top channel. After 1 hour, the AO differentiation medium was added to the top channel. In the infection experiments (Fig. 4G), the AO differentiation medium, containing either XBB.1, XBB.1.5, or Delta isolate (500 TCID$_{50}$), was inoculated into the top channel. At 2 h.p.i., the top and bottom channels were washed and cultured with AO differentiation and EGM2-MV medium, respectively. The culture supernatants were collected, and viral RNA was quantified using RT-qPCR (see "RT-qPCR" section below).

## Microfluidic device

A microfluidic device was generated according to our previous report[67,68]. Briefly, the microfluidic device consisted of two layers of microchannels separated by a semipermeable membrane. The microchannel layers were fabricated from PDMS using a soft lithographic method. PDMS prepolymer (Dow Corning, Cat# SYLGARD 184) at a base-to-curing agent ratio of 10:1 was cast against a mold composed of SU-8 2150 (MicroChem, Cat# SU-8 2150) patterns formed on a silicon wafer. The cross-sectional size of the microchannels was 1 mm in width and 330 μm in height. Access holes were punched through the PDMS using a 6-mm biopsy punch (Kai Corporation, Cat# BP-L60K) to introduce solutions into the microchannels. Two PDMS layers were bonded to a PET membrane containing 3.0-μm pores (Corning, Cat# 353091) using a thin layer of liquid PDMS prepolymer as the mortar. PDMS prepolymer was spin-coated (4000 rpm for 60 sec) onto a glass slide. Subsequently, both the top and bottom channel layers were placed on the glass slide to transfer the thin layer of PDMS prepolymer onto the embossed PDMS surfaces. The membrane was then placed onto the bottom layer and sandwiched with the top layer. The combined layers were left at room temperature for 1 day to remove air bubbles and then placed in an oven at 60 °C overnight to cure the PDMS glue. The PDMS devices were sterilized by placing them under UV light for 1 hour before cell culture.

## Preparation of human iPSC-derived lung organoids

Human iPS cell-derived lung organoids were used for Fig. 6. The iPSCl line (1383D6) (kindly provided by Dr. Masato Nakagawa, Kyoto University) was maintained on 0.5 μg/cm² recombinant human laminin 511 E8 fragments (iMatrix-511 silk, Nippi, Cat# 892021) with StemFit AK02N medium (Ajinomoto, Cat# RCAK02N) containing 10 μM Y-27632 (FUJIFILM Wako Pure Chemical, Cat# 034-24024). For passaging, iPSC colonies were treated with TrypLE Select Enzyme (Thermo Fisher Scientific, Cat# 12563029) for 10 min at 37 °C. After centrifugation, the cells were seeded onto Matrigel Growth Factor Reduced Basement Membrane (Corning, Cat# 354230)-coated cell culture plates (2.0 × 10⁵ cells/4 cm²) and cultured for 2 days. Lung organoids differentiation was performed in serum-free differentiation (SFD) medium of DMEM/F12 (3:1) (Thermo Fisher Scientific, Cat# 11320033) supplemented with N2 (FUJIFILM Wako Pure Chemical, Cat# 141-08941), B-27 Supplement Minus Vitamin A (Thermo Fisher Scientific, Cat# 12587001), ascorbic acid (50 μg/mL, STEMCELL Technologies, Cat# ST-72132), 1× GlutaMAX (Thermo Fisher Scientific, Cat# 35050-079), 1% monothioglycerol

(FUJIFILM Wako Pure Chemical, Cat# 195-15791), 0.05% bovine serum albumin, and 1× penicillin–streptomycin. For definitive endoderm induction, the cells were cultured for 3 days (days 0–3) using SFD medium supplemented with 10 μM Y-27632 and 100 ng/mL recombinant Activin A (R&D Systems, Cat# 338-AC-010). For anterior foregut endoderm induction (days 3–5), the cells were cultured in SFD medium supplemented with 1.5 μM dorsomorphin dihydrochloride (FUJIFILM Wako Pure Chemical, Cat# 047-33763) and 10 μM SB431542 (FUJIFILM Wako Pure Chemical, Cat# 037-24293) for 24 h and then in SFD medium supplemented with 10 μM SB431542 and 1 μM IWP2 (Stemolecule, Cat# 04-0034) for another 24 h. For the induction of lung progenitors (days 5–12), the resulting anterior foregut endoderm was cultured with SFD medium supplemented with 3 μM CHIR99021 (FUJIFILM Wako Pure Chemical, Cat# 032-23104), 10 ng/mL human FGF10 (PeproTech, Cat# 100-26), 10 ng/mL human FGF7 (PeproTech, Cat# 100-19), 10 ng/mL human BMP4 (PeproTech, Cat# 120-05ET), 20 ng/mL human EGF (PeproTech, Cat# AF-100-15), and all-trans retinoic acid (ATRA, Sigma-Aldrich, Cat# R2625) for 7 days. At day 12 of differentiation, the cells were dissociated and embedded in Matrigel Growth Factor Reduced Basement Membrane to generate organoids. For lung organoid maturation (days 12–30), the cells were cultured in SFD medium containing 3 μM CHIR99021, 10 ng/mL human FGF10, 10 ng/mL human FGF7, 10 ng/mL human BMP4, and 50 nM ATRA for 8 days. At day 20 of differentiation, the lung organoids were recovered from the Matrigel, and the resulting suspension of lung organoids (small free-floating clumps) was seeded onto Matrigel-coated 24-well cell culture plates. The organoids were cultured in SFD medium containing 50 nM dexamethasone (Selleck Chemicals, Cat# S1322), 0.1 mM 8-bromo-cAMP (Sigma-Aldrich, Cat# B7880), and 0.1 mM IBMX (3-isobutyl-1-methylxanthine) (FUJIFILM Wako Pure Chemical, Cat# 099-03411) for an additional 10 days.

## SARS-CoV-2 infection

One day before infection, Vero cells (10,000 cells), VeroE6/TMPRSS2 cells (10,000 cells), and Calu-3 cells (10,000 cells) were seeded into a 96-well plate. SARS-CoV-2 [1,000 $TCID_{50}$ for Vero cells (Fig. 4C and Supplementary Fig. 3B); 100 $TCID_{50}$ for VeroE6/TMPRSS2 cells (Fig. 4D, E and Supplementary Fig. 3C, Supplementary Fig. 3D)] was inoculated and incubated at 37 °C for 1 hour. The infected cells were washed, and 180 μl of culture medium was added. The culture supernatant (10 μl) was harvested at the indicated timepoints and used for RT-qPCR to quantify the viral RNA copy number (see **RT-qPCR** section below). The infection experiments using an airway-on-a-chip system (Fig. 4G) were performed as described above (see "Airways-on-a-chip" section).

## RT-qPCR

RT-qPCR was performed as previously described[1–5,21,22,24,29,30]. Briefly, 5 μl culture supernatant was mixed with 5 μl of 2 × RNA lysis buffer [2% Triton X-100 (Nacalai Tesque, Cat# 35501-15), 50 mM KCl, 100 mM Tris-HCl (pH 7.4), 40% glycerol, 0.8 U/μl recombinant RNase inhibitor (Takara, Cat# 2313B)] and incubated at room temperature for 10 min. RNase-free water (90 μl) was added, and the diluted sample (2.5 μl) was used as the template for real-time RT-PCR performed according to the manufacturer's protocol using a One Step TB Green PrimeScript PLUS RT-PCR kit (Takara, Cat# RR096A) and the following primers: Forward *N*, 5′-AGC CTC TTC TCG TTC CTC ATC AC-3′; and Reverse *N*, 5′-CCG CCA TTG CCA GCC ATT C-3′. The viral RNA copy number was standardized with a SARS-CoV-2 direct detection RT-qPCR kit (Takara, Cat# RC300A). Fluorescent signals were acquired using a QuantStudio 1 Real-Time PCR system (Thermo Fisher Scientific), QuantStudio 3 Real-Time PCR system (Thermo Fisher Scientific), QuantStudio 5 Real-Time PCR system (Thermo Fisher Scientific), StepOne Plus Real-Time PCR system (Thermo Fisher Scientific), CFX Connect Real-Time PCR Detection system (Bio-Rad), Eco Real-Time PCR System (Illumina),

qTOWER3 G Real-Time System (Analytik Jena), Thermal Cycler Dice Real Time System III (Takara), or 7500 Real-Time PCR System (Thermo Fisher Scientific).

## Animal experiments

Animal experiments (Fig. 5, Fig. 7, and Supplementary Fig. 4) were performed as previously described[1–5,21,22,24,29,30]. Syrian hamsters (male, 4 weeks old) were purchased from Japan SLC Inc. (Shizuoka, Japan). For the virus infection experiments, hamsters were anesthetized by intramuscular injection of a mixture of 0.15 mg/kg medetomidine hydrochloride (Domitor®, Nippon Zenyaku Kogyo), 2.0 mg/kg midazolam (Dormicum®, FUJIFILM WAKO, Cat# 135-13791) and 2.5 mg/kg butorphanol (Vetorphale®, Meiji Seika Pharma), or 0.15 mg/kg medetomidine hydrochloride, 4.0 mg/kg alfaxalone (Alfaxan®, Jurox) and 2.5 mg/kg butorphanol. XBB.1, XBB.1.5, Delta, and four recombinant viruses (10,000 $TCID_{50}$ in 100 μl) or saline (100 μl) were intranasally inoculated under anesthesia. Oral swabs were collected at the indicated timepoints. Body weight was recorded daily by 7 d.p.i. Enhanced pause (Penh) and the ratio of time to peak expiratory follow relative to the total expiratory time (Rpef) were measured every day until 7 d.p.i. of the XBB.1-, XBB.1.5-, and Delta-infected hamsters (see below). Lung tissues were anatomically collected at 2 and 5 d.p.i. The viral RNA load in the oral swabs and respiratory tissues was determined by RT-qPCR. These tissues were also used for IHC and histopathological analyses (see below).

## Lung function test

Lung function tests (Fig. 5A) were routinely performed as previously described[1–5]. The two respiratory parameters (Penh and Rpef) were measured using a Buxco Small Animal Whole Body Plethysmography system (DSI) according to the manufacturer's instructions. In brief, a hamster was placed in an unrestrained plethysmography chamber and allowed to acclimatize for 30 sec. Then, data were acquired over a 2.5-minute period by using FinePointe Station and Review software v2.9.2.12849 (DSI).

## Immunohistochemistry

Immunohistochemistry (IHC) (Figs. 5C and 7D) was performed as previously described using an Autostainer Link 48 (Dako). The deparaffinized sections were exposed to EnVision FLEX target retrieval solution high pH (Agilent, Cat# K8004) for 20 minutes at 97 °C for activation, and a mouse anti-SARS-CoV-2 N monoclonal antibody (clone 1035111, R&D Systems, Cat# MAB10474-SP, 1:400) was used as a primary antibody. The sections were sensitized using EnVision FLEX for 15 minutes and visualized by peroxidase-based enzymatic reaction with 3,3′-diaminobenzidine tetrahydrochloride (Dako, Cat# DM827) as substrate for 5 minutes. The N-protein positivity was evaluated by certificated pathologists as previously described. Images were incorporated as virtual slides by NDP.scan software v3.2.4 (Hamamatsu Photonics). The area of N-protein positivity was measured using Fiji software v2.2.0 (ImageJ).

## H&E staining

H&E staining (Figs. 5D, 7E, Fig. Supplementary Fig. 4B, C) was performed as previously described. Briefly, excised animal tissues were fixed with 10% formalin-neutral buffer solution and processed for paraffin embedding. The paraffin blocks were sectioned at a thickness of 3 μm and then mounted on MAS-GP-coated glass slides (Matsunami Glass, Cat# S9901). H&E staining was performed according to a standard protocol.

## Histopathological scoring

Histopathological scoring (Figs. 5E and 7F) was performed as previously described[1–5,21,22,24,29,30]. Pathological features—including (i) bronchitis or bronchiolitis, (ii) hemorrhage with congestive edema,

(iii) alveolar damage with epithelial apoptosis and macrophage infiltration, (iv) hyperplasia of type II pneumocytes, and (v) the area of hyperplasia of large type II pneumocytes—were evaluated by certified pathologists, and the degree of these pathological findings was arbitrarily scored using a four-tiered system as 0 (negative), 1 (weak), 2 (moderate), and 3 (severe). Type II pneumocytes function as preventing the alveoli from collapsing due to surface tension upon infection with SARS-CoV-2. "Large type II pneumocytes" are type II pneumocytes with hyperplasia exhibiting nuclei more than 10 μm in diameter. We described "large type II pneumocytes" as one of the notable histopathological features of SARS-CoV-2 infection in our previous studies. The total histological score is the sum of these five indices.

### FACS analysis of human MHC (human leukocyte antigen, HLA)-I expression

Single-cell suspensions of the human iPSC-derived lung organoids were treated with 1 × Permeabilization Buffer (e-Bioscience, Thermo Fisher Scientific, Cat# 00-8333-56), and then incubated with APC-labeled anti-human HLA class 1 (R&D systems, Cat# FAB7098A) and anti-SARS-CoV-2 NP antibodies (BIO Vision, Cat# A2061-50), followed by anti-Rabbit IgG (H + L) cross-adsorbed secondary antibody, Alexa Fluor 488 (Thermo Fisher Scientific, Cat# A-11008). Analysis was performed using a MACSQuant Analyzer (Miltenyi Biotec) and FlowJo software Ver10.9.

### Statistics and reproducibility

Statistical significance was tested using a two-sided Mann–Whitney $U$ test, a two-sided Student's $t$ test, a two-sided Welch's $t$ test, or a two-sided paired $t$-test, unless otherwise noted. The tests above were performed using Prism 9 software v9.1.1 (GraphPad Software).

In the time-course experiments (Figs. 4, 5, 7, and Supplementary Fig. 3), a multiple regression analysis including experimental conditions (i.e., the types of infected viruses) as explanatory variables and timepoints as qualitative control variables was performed to evaluate the difference between experimental conditions through all timepoints. The initial time point was removed from the analysis. The $P$ value was calculated by a two-sided Wald test. Subsequently, family-wise error rates (FWERs) were calculated by the Holm method. These analyses were performed on R v4.1.2 (https://www.r-project.org/).

Principal component analysis to represent the antigenicity of the S proteins was performed (Fig. 2). The NT50 values for biological replicates were scaled, and subsequently principal component analysis was performed using the prcomp function in R v4.1.2 (https://www.r-project.org/).

In Figs. 5C, 5D, 7D, E, Supplementary Fig. 4B, C, photographs shown are the representative areas of at least two independent experiments using four hamsters at each timepoint.

### Reporting summary

Further information on research design is available in the Nature Portfolio Reporting Summary linked to this article.

## Data availability

Surveillance datasets of SARS-CoV-2 isolates used in this study are available from the GISAID database (https://www.gisaid.org; EPI_SET_231124cy). The supplemental tables for the GISAID datasets are available in the GitHub repository (https://github.com/TheSatoLab/Omicron_XBB.1.5). The atomic coordinates and cryo-EM maps for the structures of the XBB.1.5 S protein closed-1 state (PDB code: 8JYK, EMDB code: 36724), closed-2 state (PDB code: 8JYM, EMDB code: 36726), in complex with hACE2 RBD one-up state (PDB code: 8JYN, EMDB code: 36727), in complex with hACE2 RBD two-up state (PDB code: 8JYO, EMDB code: 36728), and XBB.1.5 S RBD–hACE2 (PDB code: 8JYP, EMDB code: 36729) have been deposited in the Protein Data Bank

(www.rcsb.org) and Electron Microscopy Data Bank (www.ebi.ac.uk/emdb/). Source data are provided with this paper.

## Code availability

The computational codes used in the present study are available in the GitHub repository (https://github.com/TheSatoLab/Omicron_XBB.1.5).

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

## Acknowledgements

We would like to thank all members of The Genotype to Phenotype Japan (G2P-Japan) Consortium. We thank Ms. Haruko Kubo, Ms. Mizuho Tetsuka, and Ms. Saya Shimamura for their secretory work. We also thank Dr. Jin Kuramochi (Interpark Kuramochi Clinic, Japan) for providing patient sera. We would like to thank Dr. Kenzo Tokunaga (National Institute for Infectious Diseases, Japan), Dr. Jin Gohda (The University of Tokyo, Japan), and Dr. Masato Nakagawa (Kyoto University) for providing reagents. We thank Tokyo Metropolitan Institute of Public Health (Tokyo, Japan) for providing a clinical isolate of XBB.1.5 (strain TKYmbc30523/2022) and all members of the Japanese Consortium on Structural Virology (JX-Vir). We appreciate the technical assistance from The Research Support Center, Research Center for Human Disease Modeling, Kyushu University Graduate School of Medical Sciences. We gratefully acknowledge all data contributors, i.e., the authors and their originating laboratories responsible for obtaining the specimens, and their submitting laboratories for generating the genetic sequence and metadata and sharing via the GISAID Initiative, on which this research is based. The super-computing resource was provided by the Human Genome Center at The University of Tokyo. We thank Catherine Perfect, MA (Cantab), from Edanz (https://jp.edanz.com/ac), for editing a draft of this manuscript. This study was supported in part by AMED SCARDA Japan Initiative for World-leading Vaccine Research and Development Centers "UTOPIA" (JP223fa627001, to Kei Sato), AMED SCARDA Program on R&D of new generation vaccine including new modality application (JP223fa727002, to Kei Sato); AMED SCARDA Kyoto University Immunomonitoring Center (KIC) (JP223fa627009, to Takao Hashiguchi); AMED SCARDA Hokkaido University Institute for Vaccine Research and Development (HU-IVReD) (JP223fa627005, to Katsumi Maenaka; 223fa627005h0001, to Takasuke Fukuhara, and Keita Matsuno); AMED Project for Advanced Drug Discovery and Development (JP21nf0101627, to Takasuke Fukuhara); AMED Research Program on Emerging and Re-emerging Infectious Diseases (JP21fk0108574, to Hesham Nasser; JP21fk0108465, to Akatsuki Saito; JP21fk0108493, to Takasuke Fukuhara; JP22fk0108617, to Takasuke Fukuhara; JP22fk0108516, to Takasuke Fukuhara; JP22fk0108146, to Kei Sato; JP21fk0108494 to G2P-Japan Consortium, Keita Matsuno, Shinya Tanaka, Terumasa Ikeda, Takasuke Fukuhara, Takashi Irie, and Kei Sato; JP22fk0108506, to Kazuo Takayama, Akatsuki Saito, and Kei Sato; JP22fk0108534 to Takashi Irie); AMED Research Program on HIV/AIDS (JP22fk0410033, to Akatsuki Saito; JP22fk0410047, to Akatsuki Saito; JP22fk0410055, to Terumasa Ikeda; and JP22fk0410039, to Kei Sato); AMED CRDF Global Grant (JP22jk0210039 to Akatsuki Saito); AMED Japan Program for Infectious Diseases Research and Infrastructure (JP22wm0325009, to Akatsuki Saito; JP22wm0125008 to Keita Matsuno); AMED CREST (JP21gm1610005, to Kazuo Takayama; JP22gm1610008, to Takasuke Fukuhara; JP22gm1810004, to Katsumi Maenaka); JST PRESTO (JPMJPR22R1, to Jumpei Ito); JST CREST (JPMJCR20H4, to Kei Sato; JPMJCR20H8, to Takao Hashiguchi); JSPS KAKENHI Fund for the Promotion of Joint International Research (International Leading Research) (JP23K20041 to Kanako Terakado Kimura, Akatsuki Saito, Keita Matsuno, Kei Sato, and Takasuke Fukuhara); JSPS KAKENHI Grant-in-Aid for Scientific Research C (22K07103, to Terumasa Ikeda); JSPS KAKENHI Grant-in-Aid for Scientific Research B (21H02736, to Takasuke Fukuhara); JSPS KAKENHI Grant-in-Aid for Early-Career Scientists (22K16375, to Hesham Nasser; 20K15767, to Jumpei Ito); JSPS KAKENHI grant 20H05773 (to Takao Hashiguchi) and JP20H05873 (to Katsumi Maenaka); JSPS Core-to-Core Program (A. Advanced Research Networks) (JPJSCCA20190008, to Kei Sato); JSPS Research Fellow DC2 (22J11578, to Keiya Uriu); JSPS Leading Initiative for Excellent Young Researchers (LEADER) (to Terumasa Ikeda); World-leading Innovative and Smart Education (WISE) Program 1801 from the Ministry of Education, Culture, Sports, Science and Technology (MEXT) (to Naganori Nao); Ministry of Health, Labour and Welfare (MHLW) under grant 23HA2010 (to Naganori Nao and Keita Matsuno); Research Support Project for Life Science and Drug Discovery [Basis for Supporting Innovative Drug Discovery and Life Science Research (BINDS)] from AMED under the Grant JP22ama121001 (to Takao Hashiguchi) and JP22ama121037 (to Katsumi Maenaka); The Cooperative Research Program (Joint Usage/Research Center program) of Institute for Life and Medical Sciences, Kyoto University (to Kei Sato and Katsumi Maenaka); Hirose Foundation (to Tomokazu Tamura); The Tokyo Biochemical Research Foundation (to Kei Sato); Takeda Science Foundation (to Takasuke Fukuhara, Terumasa Ikeda, and Katsumi Maenaka); Hokkaido University (to Tomokazu Tamura); Hokkaido University Support Program for Frontier Research (to Takasuke Fukuhara); Mochida Memorial Foundation for Medical and Pharmaceutical Research (to Terumasa Ikeda); The Naito Foundation (to Terumasa Ikeda); Shin-Nihon Foundation of Advanced Medical Research (to Terumasa Ikeda); Waksman Foundation of Japan (to Terumasa Ikeda); an intramural grant from Kumamoto University COVID-19 Research Projects (AMABIE) (to Terumasa Ikeda); International Joint Research Project of the Institute of Medical Science, the University of Tokyo (to Terumasa Ikeda, Takasuke Fukuhara, and Akatsuki Saito); and Tsuchiya Mitsubishi Foundation (to Takashi Irie and Kei Sato).

## Author contributions

Arnon Plianchaisuk performed phylogenetic and bioinformatics analyses. Tomokazu Tamura, Takashi Irie, Sayaka Deguchi, Hesham Nasser, Keiya Uriu, Maya Shofa, Mst Monira Begum, Ryo Shimizu, Michael Jonathan, Takeshi Kondo, Hayato Ito, Akatsuki Saito, Kazuo Takayama, and Terumasa Ikeda performed cell culture experiments. Tomokazu Tamura, Keita Mizuma, Saori Suzuki, Rigel Suzuki, Takeshi Kondo, Akifumi Kamiyama, Kumiko Yoshimatsu, Naganori Nao, Keita Matsuno, and Takasuke Fukuhara performed animal experiments. Masumi Tsuda, Lei Wang, Yoshitaka Oda, and Shinya Tanaka performed histopathological analysis. Sayaka Deguchi, Rina Hashimoto, and Kazuo Takayama prepared airway-on-a-chip systems. Hisano Yajima, Yuki Anraku, Kanako Terakado Kimura, Jiei Sasaki, Kaori Sasaki-Tabata, Katsumi Maenaka, and Takao Hashiguchi performed structural analyses. Jumpei Ito performed statistical analyses. Tomokazu Tamura, Terumasa Ikeda, Akatsuki Saito, Keita Matsuno, Jumpei Ito, Shinya Tanaka, Kei Sato, Takao Hashiguchi, Kazuo Takayama, and Takasuke Fukuhara designed the experiments and interpreted the results. Tomokazu Tamura and Takasuke Fukuhara wrote the original manuscript. All authors reviewed and proofread the manuscript. The Genotype to Phenotype Japan (G2P-Japan) Consortium contributed to the project administration.

## Competing interests

The authors declare no competing interests.

## Additional information

[1]Department of Microbiology and Immunology, Faculty of Medicine, Hokkaido University, Sapporo, Japan. [2]Graduate School of Medicine, Hokkaido University, Sapporo, Japan. [3]School of Medicine, Hokkaido University, Sapporo, Japan. [4]Institute for the Advancement of Higher Education, Hokkaido University, Sapporo, Japan. [5]Institute for Vaccine Research and Development (IVReD), Hokkaido University, Sapporo, Japan. [6]One Health Research Center, Hokkaido University, Sapporo, Japan. [7]Graduate School of Biomedical and Health Sciences, Hiroshima University, Hiroshima, Japan. [8]Center for iPS Cell Research and Application (CiRA), Kyoto University, Kyoto, Japan. [9]Laboratory of Medical Virology, Institute for Life and Medical Sciences, Kyoto University, Kyoto, Japan. [10]Department of Cancer Pathology, Faculty of Medicine, Hokkaido University, Sapporo, Japan. [11]Institute for Chemical Reaction Design and Discovery (WPI-ICReDD), Hokkaido University, Sapporo, Japan. [12]Division of Molecular Virology and Genetics, Joint Research Center for Human Retrovirus Infection, Kumamoto University, Kumamoto, Japan. [13]Department of Clinical Pathology, Faculty of Medicine, Suez Canal University, Ismailia, Egypt. [14]Division of Risk Analysis and Management, International Institute for Zoonosis Control, Hokkaido University, Sapporo, Japan. [15]Division of Systems Virology, Department of Microbiology and Immunology, The Institute of Medical Science, The University of Tokyo, Tokyo, Japan. [16]Graduate School of Medicine, The University of Tokyo, Tokyo, Japan. [17]Institute for Genetic Medicine, Hokkaido University, Sapporo, Japan. [18]Department of Veterinary Science, Faculty of Agriculture, University of Miyazaki, Miyazaki, Japan. [19]Graduate School of Medicine and Veterinary Medicine, University of Miyazaki, Miyazaki, Japan. [20]Laboratory of Biomolecular Science and Center for Research and Education on Drug Discovery, Faculty of Pharmaceutical Sciences, Hokkaido University, Sapporo, Japan. [21]Department of Medicinal Sciences, Graduate School of Pharmaceutical Sciences, Kyushu University, Fukuoka, Japan. [22]Division of Pathogen Structure, International Institute for Zoonosis Control, Hokkaido University, Sapporo, Japan. [23]Global Station for Biosurfaces and Drug Discovery, Hokkaido University, Sapporo, Japan. [24]Division of International Research Promotion, International Institute for Zoonosis Control, Hokkaido University, Sapporo, Japan. [25]Center for Animal Disease Control, University of Miyazaki, Miyazaki, Japan. [26]International Collaboration Unit, International Institute for Zoonosis Control, Hokkaido University, Sapporo, Japan. [27]International Research Center for Infectious Diseases, The Institute of Medical Science, The University of Tokyo, Tokyo, Japan. [28]Graduate School of Frontier Sciences, The University of Tokyo, Kashiwa, Japan. [29]CREST, Japan Science and Technology Agency, Kawaguchi, Japan. [30]International Vaccine Design Center, The Institute of Medical Science, The University of Tokyo, Tokyo, Japan. [31]Collaboration Unit for Infection, Joint Research Center for Human Retrovirus Infection, Kumamoto University, Kumamoto, Japan. [32]Kyoto University Immunomonitoring Center, Kyoto University, Kyoto, Japan. [33]AMED-CREST, Japan Agency for Medical Research and Development (AMED), Tokyo, Japan. [34]Laboratory of Virus Control, Research Institute for Microbial Diseases, Osaka University, Suita, Japan. [41]These authors contributed equally: Tomokazu Tamura, Takashi Irie, Sayaka Deguchi, Hisano Yajima, Masumi Tsuda, Hesham Nasser, Keita Mizuma, Arnon Plianchaisuk, Saori Suzuki. ✉e-mail: tanaka@med.hokudai.ac.jp; keisato@g.ecc.u-tokyo.ac.jp; hashiguchi.takao.1a@kyoto-u.ac.jp; kazuo.takayama@cira.kyoto-u.ac.jp; fukut@pop.med.hokudai.ac.jp

## The Genotype to Phenotype Japan (G2P-Japan) Consortium

Hirofumi Sawa[5,6,24,35], Ryoko Kawabata[7], Yukio Watanabe[8], Ayaka Sakamoto[8], Naoko Yasuhara[8], Tateki Suzuki[9], Yukari Nakajima[9], Zannatul Ferdous[10], Kenji Shishido[10], Yuka Mugita[12], Otowa Takahashi[12], Kimiko Ichihara[12], Yu Kaku[15], Naoko Misawa[15], Ziyi Guo[15], Alfredo Hinay Jr.[15], Yusuke Kosugi[15,16], Shigeru Fujita[15,16], Jarel M. Tolentino[15,27], Luo Chen[15,16], Lin Pan[15,27], Mai Suganami[15], Mika Chiba[15], Ryo Yoshimura[15], Kyoko Yasuda[15], Keiko Iida[15], Naomi Ohsumi[15], Adam P. Strange[15], Yuki Shibatani[18], Tomoko Nishiuchi[18], Shiho Tanaka[15], Olivia Putri[15], Gustav Joas[15], Yoonjin Kim[15], Daichi Yamasoba[15,36], Kazuhisa Yoshimura[37], Kenji Sadamasu[37], Mami Nagashima[37], Hiroyuki Asakura[37], Isao Yoshida[37], So Nakagawa[38], Akifumi Takaori-Kondo[39], Kotaro Shirakawa[39], Kayoko Nagata[39], Ryosuke Nomura[39], Yoshihito Horisawa[39], Yusuke Tashiro[39], Yugo Kawai[39], Takamasa Ueno[40], Chihiro Motozono[40] & Mako Toyoda[40]

[35]Division of Molecular Pathobiology, International Institute for Zoonosis Control, Hokkaido University, Sapporo, Japan. [36]School of Medicine, Kobe University, Kobe, Japan. [37]Tokyo Metropolitan Institute of Public Health, Tokyo, Japan. [38]Tokai University School of Medicine, Isehara, Japan. [39]Kyoto University, Kyoto, Japan. [40]Division of Infection and Immunity, Joint Research Center for Human Retrovirus Infection, Kumamoto University, Kumamoto, Japan.

