## [Peer Review File · Nature Communications]

Reviewers' Comments:

Reviewer #1:

Remarks to the Author:

The article "Virological characteristics of the SARS-CoV-2 XBB.1.5 variant" from Tamura et al., is reporting a comparative overview of the antigenic and pathogenic phenotype of XBB.1.5 in comparison to XBB.1.

The article gives important new insights into the characteristics of this newly emerged VOI and is written concise and clear.

However, I would like to raise some specific comments:

II.95: "these data suggest that the mutations in ORF8 and S could enhance spreading of XBB.1.5 in humans." This is a strong statement which is not underlined with results that are presented in the abstract. Here you just discuss a comparable structure and decreased pathology. Why is that enhancing spread?

II.121: Do you really have concrete evidence that changes in ACE2 binding affinity correlate with enhanced transmission? For my understanding, these are different characteristics that are not necessarily linked to each other.

II.175: I guess you meant to refer to Figure S1.

II.218: Why is the fusogenicity compared to Omicron BA.2, but growth kinetics to Delta? More consistency in the comparison strategy would underline the message and be more convincing.

II. 245-253: Also include a brief description of results for Delta and compare it to those (consistent with previous chapter).

II.261: "XBB.1.5 infection resulted in increased impairment of the dynamics of weight change compared with XBB.1...". However, in Figure 5A, XBB.1 induced more BW loss.

II.271: Although you define it already as "slightly lower" pathogenicity, the differences shown in Fig.5A are very marginal. Be careful not to overinterpret these slight changes.

II.309: You only describe differences observed 5 dpi. What about 2 dpi? There you the opposite. Is it maybe just a difference in infection kinetics?

I.329: You mean similar between XBB.1 and Delta?

II.318-331: Do you also want to refer to Fig. S3? At the moment it is not referred to in the main text.

II. 353: Which data do indicate that they act in opposition? You indicate that the 486 S mutation is enhancing pathology, but direct evidence for that does not become clear.

I.415: "A smaller reduction of body weight", compared to?

Figure 4 C-E: Do you observe the same trend looking at infectious viral titers compared to RNA?

Figure 4F: What size are the size bars?

Figure 4G: It is slightly confusing in the right panel that it is referred to top and bottom. From the figure legend I understand that it is the ratio of both? How does the left panel look at 7 or 10 dpi, because it seems as if XBBs and Delta are almost getting equal towards the end of the curve.

Figure 5B: You describe the enhanced transmissibility of XBB.1.5, but in oral swabs even less RNA load is detected at 5dpi compared to Delta and XBB.1. How do you explain that?

Figure 5E: Explain what is meant with "Alveolar type II pneumocytes"? Could be antigen presence, dysmorph? Although XBB.1 stays in the bronchioles, you do observe more alveolar damage. Can you explain that via different extend of immunopathogenesis? Did you look into that?

ll.845: Don't you want to refer to Fig.6?

Figure 7A: Since you induce changes into the XBB.1 backbone, the statistical comparison not only to rXBB.1.5 but also to XBB.1 would be useful to include. How do you explain that the BW loss is higher with the recombinant viruses than with wt viruses?

Figure 7D: How valid is your comparison? IHC 2 dpi of rXBB1.5 is not resembling what you observe with wt virus.

Figure 7E: Is there a quantification available such as in Fig.S2? Would be more convincing for this comparative approach. Generally, there seems to be more and more severe inflammation compared to Figure 5. Can you explain that?

Reviewer #2:

Remarks to the Author:

In this article, this consortium of authors reports an in-depth characterization assessment of a new variant of interest of SARS-CoV-2 virus, XBB.1.5, as it compares to a closely related XBB.1 virus and the previously well-studied Delta variant; this appears to be a follow-up to a study published a few months ago, PMID 36736338. Authors conclude that XBB.1 and XBB.1.5 have similar capacities with regard to immune escape, structure, membrane fusion ability, and replication. Authors do conclude that XBB.1.5 is less pathogenic than XBB.1 but as discussed in comments below, this conclusion appears rather tenuous. Collectively, this study adds to a growing body of work from this collaborative group which runs a new variant of interest through a well-established panel of tests to publish rapid assessments; these studies are often rather similar to each other but are nonetheless helpful and important to the field.

Major comments:

1. Why did authors quantify virus in both in vitro (Fig 4C-E, G) and in vivo (Fig 5B, 7B) assays using RNA and not infectious virus as the readout? Infectious virus would be more relevant and applicable towards translating results from these laboratory experiments to humans.

2. Figure 4F/lines 238-244, authors use images of cell monolayers 48 and 72 hrs p.i. to conclude that XBB.1.5 causes cytopathic effect "more quickly and severely than XBB.1", but this conclusion cannot be drawn from the images shown in Figure 4F, for which qualitative conclusions only can be supported. Authors must either provide additional experimentation (e.g. cell death-specific assays) if they wish to state that a meaningful difference in cell death is occurring between these two viruses post-infection, or the authors must temper this statement accordingly, because the images shown in Figure 4F are not sufficient to support this statement in the text in its current form, and rather support that XBB.1 and XBB.5 are generally similar in this regard.

3. In vivo hamster data shown in Figure 5A and B: the authors are doing their best to write text supporting these figures that draws the conclusion that XBB.1.5 is less pathogenic than XBB.1 in hamsters, but to my eyes these viruses are essentially indistinguishable and cause generally similar outcomes across all measures assessed. While there are weight loss differences shown between both groups in Fig 5A, the maximum weight loss is still less than 5% in both groups, making these differences essentially negligible and not meaningful. Authors specifically call out places where viral RNA copies are higher in the lung tissues following XBB.1.5 virus infection but do not, for example, acknowledge in the text that XBB.1 is higher in oral swabs. Furthermore, for all three metrics in Figure 5B, differences between XBB.1 and XBB.1.5 are within one log difference, which again makes for a very subtle difference between these groups. In the absence of infectious virus detection (which might show more striking differences if the authors had used this more relevant readout), it seems a stretch to draw the conclusions the authors are endeavoring to do (e.g. page 8 lines 270-72, page 12 lines 414-415) in the text. Similarly, I do not understand how the authors are drawing the conclusion (page 9 lines 315-6, page 12 lines 417-18) that XBB.1.5 has diminished pathogenicity relative to XBB.1 based on the data shown in Figure 5E, for similar reasons.

4. In the abstract (page 3 lines 95-6) the authors state that "...the mutations in ORF8 and S could

enhance spreading of XBB.1.5 in humans” but no virus transmissibility data is presented in this study. As such, the authors should be more precise in their wording here to more clearly specify what data presented in this study supports (evasion of immune response, vaccine responses, etc) and not transmission.

Minor comments:

1. Page 5 lines 133-4, this weblink could be moved into the references section.
2. Page 5 lines 160-61, how many doses were given of the BA.1 and BA.5 bivalent vaccines prior to collection of serum?
3. Page 7 line 220, specify “Calu-3/DSP1-7” cells here to match what is listed in the legend and state why these cells were employed in lieu of standard Calu-3 cells.
4. Page 7 lines 232-234, the text refers to work performed at an MOI=0.01 which is 4C and 4D, so 4E (MOI=0.1) should not be specified on line 234.
5. Check the statistics listed in Figure 4D, they appear to be inverted in the figure.
6. Page 14 lines 452-3, no star appears on Figure 1, only diamonds to identify mutations of interest.
7. Page 14 line 462, authors state that geometric mean and CI are shown in the graphs but the coloration of these lines is identical to the data points so the reader cannot see them, please recolor this summary data in black so it can stand out from the data itself.

Reviewer #3:

Remarks to the Author:

The authors characterize the SARS-CoV-2 XBB.1.5 variant, which has become dominant recently, and find minor differences to the XBB.1 variant, on top of the previously reported ones. The S protein of XBB.1.5, bearing the F486P mutation, was reconstructed in its apo and ACE2-bound forms and compared to the previously reconstructed XBB.1 S. No major differences were found, except in the L828 to Q836 mobile loop.

Technically, the cryo-EM data collection and processing were done well. The results are not over-interpreted and the methods are detailed, which is nice. I have no major comments otherwise.

Reviewer #4:

Remarks to the Author:

The main goal of the paper is to investigate the evolution SARS-CoV-2 XBB.1.5 variant and characterize its pathogenicity and transmissibility compared to XBB.1, which is another omicron sub-variant. The authors describe a number of in vitro assays, as well as hamsters model experiments showing that XBB.1.5 is intrinsically less pathogenic than XBB.1, and that its mutations are linked to reduced virulence.

I found the experimental results fairly convincing and relying on solid methodology. However, the general finding of XBB.1.5 decreased virulence and enhanced transmissibility is not surprising and provides limited incremental insights. Most importantly, the conclusion that XBB.1.5 evolved from XBB.1, albeit reasonable, is completely unsupported by the phylogenetic analysis, as it is the reconstruction of ancestral mutations evolved along B.1.5 phylogenetic lineage.

Major Remarks:

1. The sequence dataset assembled from the GISAID database for evolutionary analysis is problematic. Randomly downsampling million of sequences in the database, instead of using an

appropriate sampler minimizing bias (e.g., the one in Marini et al. 2022), is not a good strategy, given GISAID well known bias due to overrepresentation of sequences from UK, US and a few other countries. Contrary to what the authors claim, their small dataset is not at all representing the full extent of SARS-CoV-2 genetic diversity and any analysis is guaranteed to be biased as well.

2. The phylogenetic analysis is very superficial. The ML tree is essentially unresolved, as shown by the lack of any high bootstrap value along internal branches. In particular, bootstrap values < 50% for the XBB sub-clades show that more than half of the replicated phylogenies do not include those sub-clades, making them completely unreliable and phylogenetically meaningless.

3. The ancestral reconstruction of the mutations leading to the XB.1.5 variant is also unreliable, due to sampling bias in the data (see above) and because the method does not take into account evolutionary relationships.

Answers to Reviewer #1

The article "Virological characteristics of the SARS-CoV-2 XBB.1.5 variant" from Tamura et al., is reporting a comparative overview of the antigenic and pathogenic phenotype of XBB.1.5 in comparison to XBB.1.

The article gives important new insights into the characteristics of this newly emerged VOI and is written concise and clear.

However, I would like to raise some specific comments:

We thank this reviewer very much for his/her/their time to evaluate our manuscript. We will answer all your concerns one-by-one. Please find our revised manuscript in line with your comments.

Comments:

ll.95: "these data suggest that the mutations in ORF8 and S could enhance spreading of XBB.1.5 in humans." This is a strong statement which is not underlined with results that are presented in the abstract. Here you just discuss a comparable structure and decreased pathology. Why is that enhancing spread?

As XBB.1.5 is outcompeted to the ancestral XBB.1 in human population and have been acquired two mutations in spike and ORF8 as shown in **Fig. 1** and **Fig. S1**, suggesting these mutations are responsible for viral spreading in humans. By our comprehensive analysis in this study and the previous one, acquisition of the S:F486P substitution confers augmented ACE2 binding affinity without losing immune resistance (**Fig. 2**), which leads to greater transmissibility. Despite our best efforts to characterize XBB.1.5 *in vitro* and *in vivo*, and also three-dimensional structure of Spike and hACE2 complex, there are still missing pieces left toward understanding of whole picture "why SARS-CoV-2 is evolving" and "XBB.1.5 is outcompeted to the ancestral XBB.1". In conclusion, two reviewers (Reviewer #1 and Reviewer #2) are concern about our claim described in the initial manuscript, we tone down the conclusion and revised accordingly.

Lines 96-97 in the revision:

Together, our study identified the two viral functions defined the difference between XBB.1 and XBB.1.5.

Lines 122-124 in the revision:

This suggested that the XBB.1.5 acquired S:S486P substitution which confers augmented ACE2 binding affinity without losing immune resistance leading to greater transmissibility compared to XBB.1.

Line 452 in the revision:

..., XBB.1.5 may have acquired...

ll.121: Do you really have concrete evidence that changes in ACE2 binding affinity correlate with enhanced transmission? For my understanding, these are different characteristics that are not necessarily linked to each other.

Thank you for pointing it out. In general, hACE2 binding affinity is not the sole determinant of enhanced transmission. In this study, we compared the XBB.1 and XBB.1.5. Compared to XBB.1, XBB.1.5 acquired the single mutation in S, and XBB.1.5 S binds to hACE2 more efficiently than XBB.1 S, suggesting that S486P contributes greater transmissibility of XBB.1.5 than XBB.1. The text was revised in a manner that does not generalize the correlation between hACE2 binding affinity and transmissibility.

Lines 122-124 in the revision:

This suggested that the XBB.1.5 acquired S:S486P substitution which confers augmented ACE2 binding affinity without losing immune resistance leading to greater transmissibility compared to XBB.1.

ll.175: I guess you meant to refer to Figure S1.

Thank you for carefully reading out. We amended it accordingly.

Lines 178 in the revision:

...(Fig. 3A, Fig. S2, and Table S3).

ll.218: Why is the fusogenicity compared to Omicron BA.2, but growth kinetics to Delta? More consistency in the comparison strategy would underline the message and be more convincing.

Thank you for the note. We added the data of fusogenicity of Delta S protein in the revised manuscript. Because the fusogenic ability of Delta exceeded to the other Omicron subvariants, we put the data of fusogenic profile in a separate panel; Fig. 4B in the revision is same one in the initial submission and new Fig. S3A in this version is compile all of the data in the single bar graph. The text was revised accordingly.

Lines 230-231 in the revision:

(Fig. 4A and Fig. S3A). As expected, the fusogenic ability of Delta S was greatest among we examined (Fig. S3A).

Lines: 1380-1381 in the revision:

(A) (Left) Fusion activity (arbitrary units) of the Delta, BA.2, XBB.1, and XBB.1.5 S proteins are shown.

II. 245-253: Also include a brief description of results for Delta and compare it to those (consistent with previous chapter).

Thank you. Upon the reviewer's request, we added a description about Delta regarding experiments using airway-on-a-chip.

Lines 265-269 in the revision:

However, the barrier-disrupting capacity of XBB.1.5 was significantly lower than that of Delta.

II.261: "XBB.1.5 infection resulted in increased impairment of the dynamics of weight change compared with XBB.1...". However, in Figure 5A, XBB.1 induced more BW loss. We are sorry for the reviewer to make a confusion. The manuscript is revised accordingly.

Lines 275-277 in the revision:

The hamsters infected with XBB.1 exhibited statistically more reduction of weight compared with those infected with XBB.1.5.

II.271: Although you define it already as "slightly lower" pathogenicity, the differences shown in Fig.5A are very marginal. Be careful not to overinterpret these slight changes.

Thank you for the note. We decided to delete this sentence.

II.309: You only describe differences observed 5 dpi. What about 2 dpi? There you the opposite. Is it maybe just a difference in infection kinetics?

As you pointed out, we described the data regarding at 2 dpi. Yes, we consider the infection kinetics is different between XBB.1 and XBB.1.5 in hamsters. The manuscript was revised accordingly.

Line 324 in the revision:

...at 5 d.p.i. (Fig. 5E).

Lines 326-328 in the revision:

... of XBB.1 was higher than that of XBB.1 at 2 d.p.i. and eventually become comparable between XBB.1.5 and XBB.1 at 5 d.p.i.. The inflammation...

I.329: You mean similar between XBB.1 and Delta?

Yes. We revised accordingly.

Line 346 in the revision:

... the HLA-1 expression levels were similar between XBB.1 and Delta (**Fig. 6B**).

II.318-331: Do you also want to refer to **Fig. S3**? At the moment it is not referred to in the main text.

Yes, we referred to new **Fig. S5** in the revision accordingly.

Line 342 in the revision:

...or Delta variants (**Fig. 6A** and **Fig. S5**)

II. 353: Which data do indicate that they act in opposition? You indicate that the 486 S mutation is enhancing pathology, but direct evidence for that does not become clear.

Thank you very much for the note. The ORF8 dysfunction by G8 nonsense mutation contributed to impairment of immune activation as shown in **Fig. 6** (*ex vivo*) and in **Fig. 7E** and **7F** (*in vivo*). Regarding the effects of S:S486P mutation in pathogenicity, this mutation confers the spike to bind human ACE2 more efficiently as shown in **Fig. 3**, leading to enter the virus more efficiently. Because we used the hamsters instead of transgenic mice expressing hACE2, very little effects of this mutation could be evaluated upon infection with the recombinant viruses carrying each single mutation. According to her/his/their comment, we toned down the conclusion and modified the text accordingly.

Lines 371-372 in the revision:

...both mutations, S:S486P and ORF8:G8stop are involved in the difference of viral pathogenicity between XBB.1 and XBB.1.5 in hamsters.

I.415: "A smaller reduction of body weight", compared to?

Thank you. We added " to XBB.1 and Delta" accordingly.

Line 433 in the revision:

... a smaller reduction of body weight compared to XBB.1 and Delta (**Fig. 5A**).

Figure 4 C-E: Do you observe the same trend looking at infectious viral titers compared to RNA?

Upon the reviewer's request, we provided the data set showing infectious titers corresponded to **Fig. 4C-E** at the initial submission and now put them in **Fig. S3B-D** accordingly. The trend looks similar between RNA copies and infectious titers.

Lines 241-242 in the revision:

We quantified viral RNA copies in supernatants (**Fig. 4C-F**) as well as infectious titers (**Fig. S3B-D**).

Lines 1382-1384 in the revision:

(**B**, **C**, and **D**) Infectious titers of supernatants were calculated by the TCID₅₀ assay and were shown in the bar graphs.

Figure 4F: What size are the size bars?

500 μ m. We added information in the figure legend accordingly.

Line 518 in the revision:

Scale bar: 500 μ m.

Figure 4G: It is slightly confusing in the right panel that it is referred to top and bottom. From the figure legend I understand that it is the ratio of both? How does the left panel look at 7 or 10 dpi, because it seems as if XBBs and Delta are almost getting equal towards the end of the curve.

The number of viral copies in the top channel will be almost the same for all variants. However, because XBBs only partially disrupt the airway epithelial and endothelial cell barrier, the viral copy number in the bottom channel of XBBs-infected airway-on-a-chip is always lower than that of Delta-infected airway-on-a-chip.

Figure 5B: You describe the enhanced transmissibility of XBB.1.5, but in oral swabs even less RNA load is detected at 5dpi compared to Delta and XBB.1. How do you explain that?

We have two reasons to conclude that XBB.1.5 might have acquisition of enhanced

transmissibility. First, the Spike-hACE2 complex of XBB.1 and XBB.1.5 showed that binding affinity is enhanced by S486P mutation in XBB.1.5 as discussed above. Secondly, in hamsters, XBB.1.5 possesses slightly higher ability of viral shedding and dissemination compared with the ancestral XBB.1. However, because XBB.1.5 acquired the mutation to disrupt the function of ORF8 and did not suppress MHC-1 efficiently, the virus shedding impaired at the later time point.

Figure 5E: Explain what is meant with "Alveolar type II pneumocytes"? Could be antigen presence, dysmorphic? Although XBB.1 stays in the bronchioles, you do observe more alveolar damage. Can you explain that via different extend of immunopathogenesis? Did you look into that?

Type II pneumocytes secrete surfactant proteins onto the surface of the alveolar epithelium, which play an important role in preventing the alveoli from collapsing due to surface tension. Therefore, Type II pneumocytes proliferate during alveolar damage. In general, when the virus including SARS-CoV-2 is administered through the respiratory tract, bronchitis/bronchiolitis occurs first, followed by gradual alveolar damage. Subsequently, increased vascular permeability causes congestion and hemorrhage.

In our animal model, XBB.1 infection resulted in more prolonged stay in the bronchioles than XBB.1.5 infection at 2.d.p.i. (**Fig. 5C**), leading to cause severe bronchitis/bronchiolitis, followed by alveolar damage in the surrounding area (**Fig. 5D, 5E**). At the same time, Type II pneumocytes proliferated to protect or repair alveolar architecture from alveolar damage and congestion/hemorrhage. On the other hand, the number of Type II pneumocytes decreased at 5.d.p.i., because the XBB.1 almost disappeared and the alveolar damage has improved (**Fig. 5C, 5E**). We added the sentence of description of Type II pneumocyte upon infection with the viruses in the revised manuscript.

Lines 330-332 in the revision:

Consistent with this observation, the dynamics of Type II pneumocytes were opposite between these two XBBs.

ll.845: Don't you want to refer to **Fig.6**?

Thank you for the note. We used human iPSC-derived lung organoids of the experiments described in **Fig. 6**. To clarify this, we referred to **Fig. 6** in the methods section of the preparation of human iPSC-derived lung organoids.

Lines 943-986 in the revision:

Preparation of human iPSC-derived lung organoids

Human iPSC-derived lung organoids were used for **Fig. 6**. The iPSC line (1383D6) (kindly provided by Dr. Masato Nakagawa, Kyoto University) was maintained on 0.5 $\mu\text{g}/\text{cm}^2$ recombinant human laminin 511 E8 fragments (iMatrix-511 silk, Nippi, Cat# 892021) with StemFit AK02N medium (Ajinomoto, Cat# RCAF02N) containing 10 μM Y-27632 (FUJIFILM Wako Pure Chemical, Cat# 034-24024). For passaging, iPSC colonies were treated with TrypLE Select Enzyme (Thermo Fisher Scientific, Cat# 12563029) for 10 min at 37 °C. After centrifugation, the cells were seeded onto Matrigel Growth Factor Reduced Basement Membrane (Corning, Cat# 354230)-coated cell culture plates (2.0×10^5 cells/4 cm^2) and cultured for 2 days. Lung organoids differentiation was performed in serum-free differentiation (SFD) medium of DMEM/F12 (3:1) (Thermo Fisher Scientific, Cat# 11320033) supplemented with N2 (FUJIFILM Wako Pure Chemical, Cat# 141-08941), B-27 Supplement Minus Vitamin A (Thermo Fisher Scientific, Cat# 12587001), ascorbic acid (50 $\mu\text{g}/\text{mL}$, STEMCELL Technologies, Cat# ST-72132), 1 \times GlutaMAX (Thermo Fisher Scientific, Cat# 35050-079), 1% monothioglycerol (FUJIFILM Wako Pure Chemical, Cat# 195-15791), 0.05% bovine serum albumin, and 1 \times penicillin–streptomycin. For definitive endoderm induction, the cells were cultured for 3 days (days 0–3) using SFD medium supplemented with 10 μM Y-27632 and 100 ng/mL recombinant Activin A (R&D Systems, Cat# 338-AC-010). For anterior foregut endoderm induction (days 3–5), the cells were cultured in SFD medium supplemented with 1.5 μM dorsomorphin dihydrochloride (FUJIFILM Wako Pure Chemical, Cat# 047-33763) and 10 μM SB431542 (FUJIFILM Wako Pure Chemical, Cat# 037-24293) for 24 h and then in SFD medium supplemented with 10 μM SB431542 and 1 μM IWP2 (REPROCELL) for another 24 h. For the induction of lung progenitors (days 5–12), the resulting anterior foregut endoderm was cultured with SFD medium supplemented with 3 μM CHIR99021 (FUJIFILM Wako Pure Chemical, Cat# 032-23104), 10 ng/mL human FGF10 (PeproTech, Cat# 100-26), 10 ng/mL human FGF7 (PeproTech, Cat# 100-19), 10 ng/mL human BMP4 (PeproTech, Cat# 120-05ET), 20 ng/mL human EGF (PeproTech, Cat# AF-100-15), and all-trans retinoic acid (ATRA, Sigma-Aldrich, Cat# R2625) for 7 days. At day 12 of differentiation, the cells were dissociated and embedded in Matrigel Growth Factor Reduced Basement Membrane to generate organoids. For lung organoid maturation (days 12–30), the cells were cultured in SFD medium containing 3 μM CHIR99021, 10 ng/mL human FGF10, 10 ng/mL human FGF7, 10 ng/mL human BMP4, and 50 nM ATRA for 8 days. At day 20 of differentiation, the lung organoids were recovered from the Matrigel, and the resulting suspension of lung organoids (small free-floating clumps) was seeded onto Matrigel-coated 24-well cell culture plates. The organoids were cultured in SFD medium

containing 50 nM dexamethasone (Selleck, Cat# S1322), 0.1 mM 8-bromo-cAMP (Sigma-Aldrich, Cat# B7880), and 0.1 mM IBMX (3-isobutyl-1-methylxanthine) (FUJIFILM Wako Pure Chemical, Cat# 099-03411) for an additional 10 days.

Figure 7A: Since you induce changes into the XBB.1 backbone, the statistical comparison not only to rXBB.1.5 but also rXBB.1 would be useful to include. How do you explain that the BW loss is higher with the recombinant viruses than with wt viruses?

Thank you for the comment and careful assessment. We analyzed the data of weight change accordingly. We would list two reasons why the dynamics of body weight were different between clinical isolates and the recombinant viruses. First, the surrounding environmental condition of the experimental setting is different. The animal experiments were conducted either at International Institute for Zoonosis Control, Hokkaido University or at Institute for Genetic Medicine, Hokkaido University in this study. The experimental environment including negative pressure and size of animal cage are different of these two bio-contaminant institutions. The animals were purchased from the same breeder, but the litter was different between the set of the two experiments. Second, the clinical isolates and the recombinant viruses were prepared in the different cells with different procedures in the different laboratory. In general, clinical isolates maintain quasi-population compared to *de novo* recombinant viruses. To further investigate the viral factors consisting of different features of XBB.1 and XBB.1.5, we generated the recombinant viruses. Although the extent of body weight loss is different, the tendency is similar.

Figure 7D: How valid is your comparison? IHC 2 dpi of rXBB1.5 is not resembling what you observe with wt virus.

We presented tissues around the main bronchi as IHC of clinically-isolated XBB.1.5-infected hamsters because it is easier to distinguish differences from Delta in virus spread to the alveoli (**Fig. 5C**). On the other hand, we displayed peripheral lung tissues containing bronchioles of hamsters infected with rXBB.1.5 (**Fig. 7D**). The same findings were observed in lung tissues infected with clinically-isolated XBB.1.5 and rXBB.1.5. A small amount of N-protein positive cells was observed in the alveolar spaces around main bronchi, but N-positive cells did not spread into the alveoli of the peripheral lungs.

Figure 7E: Is there a quantification available such as in **Fig.S2**? Would be more convincing for this comparative approach. Generally, there seems to be more and more severe inflammation compared to **Figure 5**. Can you explain that?

Thank you for your advice. Although we agree with your comments, quantitative analysis was difficult in **Fig. 7E** due to unexpected artifacts such as congestion/hemorrhage in sacrifice. These artifacts were accurately excluded in the histopathological scoring by the certified pathologists (**Fig. 7F**), and thus can be used reliably in place of quantitative analysis. In addition, the actual inflammation area is marked in blue and enlarged photos of the inflammation area and the congestion area due to sacrifice is shown below.

Answers to Reviewer #2

In this article, this consortium of authors reports an in-depth characterization assessment of a new variant of interest of SARS-CoV-2 virus, XBB.1.5, as it compares to a closely related XBB.1 virus and the previously well-studied Delta variant; this appears to be a follow-up to a study published a few months ago, PMID 36736338. Authors conclude that XBB.1 and XBB.1.5 have similar capacities with regard to immune escape, structure, membrane fusion ability, and replication. Authors do conclude that XBB.1.5 is less pathogenic than XBB.1 but as discussed in comments below, this conclusion appears rather tenuous. Collectively, this study adds to a growing body of work from this collaborative group which runs a new variant of interest through a well-established panel of tests to publish rapid assessments; these studies are often rather similar to each other but are nonetheless helpful and important to the field.

Thank you very much for the positive impression of our study including of the previous series. We have done our best to answer all concerns you evaluate our manuscript. It would be grateful if you could read our answers listed below.

Major comments:

1. Why did authors quantify virus in both *in vitro* (Fig 4C-E, G) and *in vivo* (Fig 5B, 7B) assays using RNA and not infectious virus as the readout? Infectious virus would be more relevant and applicable towards translating results from these laboratory experiments to humans.

Thank you for the comment. We understand your concern. Because recent SARS-CoV-2 studies including ours reported RNA levels as indicators of infectious viral titers without any major inconsistencies, we originally employed the quantity of RNAs in our assays. To prove the ratio of RNA:TCID₅₀ is not different for each variant including the latest XBB variant and the significant differences shown in the present study can be supported by viral titers too, we examined following samples that are available for quantification: *in vitro* (Fig. 4C-E, G) and *in vivo* (Fig. 5B). Our results clearly demonstrate that the viral particles in the cell culture supernatant or tissue could complement infectivity results. Please kindly find the data in Fig. S3B-D, F and Fig. S4A in the revision.

Lines 241-242 in the revision:

We quantified viral RNA copies in supernatants (Fig. 4C-F) as well as infectious titers (Fig. S3B-D).

Lines 297-298 in the revision:

...and the infectious titers in the lungs exhibited the similar tendency of viral RNA loads (**Fig. S4A**),...

Lines 1382-1384 in the revision:

(**B**, **C**, and **D**) Infectious titers of supernatants were calculated by the TCID₅₀ assay and were shown in the bar graphs.

Lines 1386-1388 in the revision:

(**F**) Infectious titers in the top (**left**) and bottom (**middle**) channels of an airway-on-a-chip upon infection with XBB.1, XBB.1.5, and Delta are shown.

Lines 1393-1394 in the revision:

(**A**) The viral infectious titers in the lungs were measured and calculated as TCID₅₀.

2. **Figure 4F**/lines 238-244, authors use images of cell monolayers 48 and 72 hrs p.i. to conclude that XBB.1.5 causes cytopathic effect “more quickly and severely than XBB.1”, but this conclusion cannot be drawn from the images shown in **Figure 4F**, for which qualitative conclusions only can be supported. Authors must either provide additional experimentation (e.g. cell death-specific assays) if they wish to state that a meaningful difference in cell death is occurring between these two viruses post-infection, or the authors must temper this statement accordingly, because the images shown in **Figure 4F** are not sufficient to support this statement in the text in its current form, and rather support that XBB.1 and XBB.1.5 are generally similar in this regard.

To answer this reviewer’s comment, we conducted a plaque assay and quantified the diameter of plaque size upon viral infection. The data was put in the new **Fig. S3E** and the images below. XBB.1.5 induced significantly larger plaques in Vero cells compared with XBB.1, supporting the data of *in vitro* growth kinetics (**Fig. 4F**). The text was revised accordingly.

Vero cells

VeroE6/TMPRSS2 cells

Lines 254-258 in the revision:

We also quantified plaque size upon viral infection in Vero and VeroE6/TMPRSS2 cells (**Fig. S3E**). While the size of fucuses induced by XBB.1 and XBB.1.5 was comparable in VeroE6/TMPRSS2 cells, XBB.1.5 induced significantly larger plaques in Vero cells compared with XBB.1.

Lines 753-765 in the revision:

Plaque assay

Plaque assay was performed as previously described³⁰. Briefly, 1 day before infection,

Vero cells or VeroE6/TMPRSS2 cells (100,000 cells per well) were seeded into a 24-well plate and infected with SARS-CoV-2 (0.5, 5, 50, or 500 TCID₅₀) at 37°C for 2 hours. A mounting solution containing 3% FBS and 1.5% carboxymethyl cellulose (Sigma-Aldrich, Cat#C4888-500G) was overlaid, followed by incubation at 37°C. At 3 d.p.i., the culture medium was removed, and the cells were washed with PBS three times and fixed with 4% paraformaldehyde phosphate buffer solution (Nacalai Tesque, Cat# 09154-85). The fixed cells were washed with tap water, dried, and stained with a staining solution [2% Crystal Violet (Nacalai Tesque, Cat# 09804-52) in water] for 30 minutes. The stained cells were washed with tap water and dried, and the size of the plaques was measured using Adobe Photoshop 2024 v25.0.0 (Adobe).

Lines 1384-1386 in the revision:

(E) Plaque assay. Vero cells (**left**) and VeroE6/TMPRSS2 cells (**right**) were used for the target cells. A summary of the recorded plaque diameters (20 plaques per virus) is shown as bar graphs.

3. *In vivo* hamster data shown in **Figure 5A** and **B**: the authors are doing their best to write text supporting these figures that draws the conclusion that XBB.1.5 is less pathogenic than XBB.1 in hamsters, but to my eyes these viruses are essentially indistinguishable and cause generally similar outcomes across all measures assessed. While there are weight loss differences shown between both groups in **Fig. 5A**, the maximum weight loss is still less than 5% in both groups, making these differences essentially negligible and not meaningful. Authors specifically call out places where viral RNA copies are higher in the lung tissues following XBB.1.5 virus infection but do not, for example, acknowledge in the text that XBB.1 is higher in oral swabs. Furthermore, for all three metrics in Figure 5B, differences between XBB.1 and XBB.1.5 are within one log difference, which again makes for a very subtle difference between these groups. In the absence of infectious virus detection (which might show more striking differences if the authors had used this more relevant readout), it seems a stretch to draw the conclusions the authors are endeavoring to do (e.g. page 8 lines 270-72, page 12 lines 414-415) in the text. Similarly, I do not understand how the authors are drawing the conclusion (page 9 lines 315-6, page 12 lines 417-18) that XBB.1.5 has diminished pathogenicity relative to XBB.1 based on the data shown in **Figure 5E**, for similar reasons.

Thank you very much for careful assessment of our *in vivo* data. Because two reviewers including you are raising the interruption of *in vivo* pathogenicity as concerns, we

revisited all sentences involved in and tone down the conclusion.

Lines 275-277 in the revision:

The hamsters infected with XBB.1 exhibited statistically more reduction of weight compared with those infected with XBB.1.5.

Lines 326-327 in the revision:

...of XBB.1 was higher than that of XBB.1.5 at 2 d.p.i. and eventually become comparable...

4. In the abstract (page 3 lines 95-6) the authors state that "...the mutations in ORF8 and S could enhance spreading of XBB.1.5 in humans" but no virus transmissibility data is presented in this study. As such, the authors should be more precise in their wording here to more clearly specify what data presented in this study supports (evasion of immune response, vaccine responses, etc) and not transmission.

Thank you for pointing out. This concern is also raised from the Reviewer #1. We revisited our data and did our best to revise manuscript with accurate/ fair interruption. Could you please find our response to the first and second comments of the Reviewer #1?

Minor comments:

1. Page 5 lines 133-4, this weblink could be moved into the references section.

We revised accordingly. Please find the new reference listing. The indicated weblink is now referred as #16.

2. Page 5 lines 160-61, how many doses were given of the BA.1 and BA.5 bivalent vaccines prior to collection of serum?

Three time prior to receive the BA.1 and BA.5 bivalent vaccines. Thus, the donors have received the vaccine four-times in total. Please kindly find details in **Table S2**.

3. Page 7 line 220, specify "Calu-3/DSP₁₋₇" cells here to match what is listed in the legend and state why these cells were employed in lieu of standard Calu-3 cells.

We added the description of Calu-3 cells expressing DSP₁₋₇ in the revision accordingly.

Lines 222-228 in the revision:

To quantitatively monitor S protein-mediated fusion activity, we utilized DSP (dual split

protein). DSP is composed of DSP₁₋₇ and DSP₈₋₁₁, which is a hybrid protein constituted by split *Renilla* luciferase (RL) and split green fluorescence protein (GFP)²⁰. When DSP₁₋₇ and DSP₈₋₁₁ are associated after fusion, the reconstituted split proteins produce luminescence and fluorescence. Therefore, the fusogenicity of XBB.1.5 S was measured by the SARS-CoV-2 S-based fusion assay^{1,3,4,19} using Calu-3 cells stably expressing DSP₁₋₇.

4. Page 7 lines 232-234, the text refers to work performed at an MOI=0.01 which is 4C and 4D, so 4E (MOI=0.1) should not specified on line 234.

We are sorry for the careless mistake. We revised accordingly.

Lines 242-246 in the revision:

At a multiplicity of infection (MOI) of 0.01, the multistep growth of Delta in Vero cells (**Fig. 4C**) and VeroE6/TMPRSS2 cells (**Fig. 4D**) was greater than that of XBB.1 and XBB1.5, while the growth curves of XBB.1 and XBB.1.5 were almost comparable. At an MOI of 0.1 in VeroE6/TMPRSS2 cells...

5. Check the statistics listed in Figure 4D, they appear to be inverted in the figure.

We are sorry for the careless mistake. We reanalyzed the data set and revised accordingly. Please find the revised panel in Fig. 4D.

6. Page 14 lines 452-3, no star appears on Figure 1, only diamonds to identify mutations of interest.

Again, we apologize for the mistake. We have edited the figure legend accordingly.

Lines 463-472 in the revision:

Fig. 1 | Evolutionary history of the XBB.1.5 sublineage

A representative maximum likelihood-based phylogenetic tree of SARS-CoV-2 in the XBB lineage. The XBB.1.4.1, XBB.3.1, and XBB.4.1 sublineages are included in the XBB.1.4, XBB.3, and XBB.4 lineages, respectively. Diamonds represent the occurrence of mutations of interest. Only mutation occurrences at internal nodes with at least 20 and also a half of descendant tips harboring the mutation are shown. Numbers at diamonds represent Shimodaira-Hasegawa-like approximate likelihood ratio test and ultrafast bootstrap supporting values, respectively.

7. Pag 14 line 462, authors state that geometric mean and CI are shown in the graphs

but the coloration of these lines is identical to the data points so the reader cannot see them, please recolor this summary data in black so it can stand out from the data itself.

We recolored the data and revised accordingly. Please find the new Fig. 2.

Answers to Reviewer #3 (Remarks to the Author)

The authors characterize the SARS-CoV-2 XBB.1.5 variant, which has become dominant recently, and find minor differences to the XBB.1 variant, on top of the previously reported ones. The S protein of XBB.1.5, bearing the F486P mutation, was reconstructed in its apo and ACE2-bound forms and compared to the previously reconstructed XBB.1 S. No major differences were found, except in the L828 to Q836 mobile loop.

Technically, the cryo-EM data collection and processing were done well. The results are not over-interpreted and the methods are detailed, which is nice. I have no major comments otherwise.

We are really pleased to hear this high evaluation from this reviewer. Highly appreciated.

Answers to Reviewer #4 (Remarks to the Author)

The main goal of the paper is to investigate the evolution SARS-CoV-2 XBB.1.5 variant and characterize its pathogenicity and transmissibility compared to XBB.1, which is another omicron sub-variant. The authors describe a number of in vitro assays, as well as hamsters model experiments showing that XBB.1.5 is intrinsically less pathogenic than XBB.1, and that its mutations are linked to reduced virulence.

I found the experimental results fairly convincing and relying on solid methodology. However, the general finding of XBB.1.5 decreased virulence and enhanced transmissibility is not surprising and provides limited incremental insights. Most importantly, the conclusion that XBB.1.5 evolved from XBB.1, albeit reasonable, is completely unsupported by the phylogenetic analysis, as it is the reconstruction of ancestral mutations evolved along B.1.5 phylogenetic lineage.

We appreciate your concrete assessment on our experimental results and your concern on our phylogenetic analyses. Here, we have revised our phylogenetic analysis to ensure the reliability of our claims. Please kindly find our responses below.

Major Remarks:

1. The sequence dataset assembled from the GISAID database for evolutionary analysis is problematic. Randomly down sampling million of sequences in the database, instead of using an appropriate sampler minimizing bias (e.g., the one in Marini et al. 2022), is not a good strategy, given GISAID well known bias due to overrepresentation of sequences from UK, US and a few other countries. Contrary to what the authors claim, their small dataset is not at all representing the full extent of SARS-CoV-2 genetic diversity and any analysis is guaranteed to be biased as well.

We appreciate your concern regarding the sampling bias. To address this issue, we tried to install Tardis, the down sampling program of Marini et al. (Marini et al., 2021. *Bioinformatics*. 38: 856860. doi: 10.1093/bioinformatics/btab725), mentioned by the reviewer earlier. However, we were unsuccessful due to the incompatibility of computational environment. Therefore, we performed an alternative method that samples 10 genomic sequences evenly from every combination of country and PANGO lineage, mirroring the approach used by Marini et al. This strategy resulted in 2,341 to 2,347 genomic sequences per dataset, which is approximately 7.9% of the 29,608 SARS-CoV-2 isolates in the XBB lineage. Additionally, to quantitatively assess the impact of this down sampling on our phylogenetic analysis, we reconstructed ten phylogenetic trees. The outcomes of these reconstructions are discussed in the response to the

subsequent comment.

2. The phylogenetic analysis is very superficial. The ML tree is essentially unresolved, as shown by the lack of any high bootstrap value along internal branches. Bootstrap values < 50% for the XBB sub-clades show that more than half of the replicated phylogenies do not include those sub-clades, making them completely unreliable and phylogenetically meaningless.

Thank you very much for pointing out this issue. Notably, worldwide researchers are facing the difficulty to infer a phylogeny of SARS-CoV-2 with high supportive values due to large number of genomic sequences and low parsimony informative nucleotides (Morel et al., 2021. *Mol. Biol. Evol.* 38: 1777–1791. doi: 10.1093/molbev/msaa314). Our main claim in phylogenetic analysis is that S:G252V, ORF8:G8stop, and S:F486P mutations occurred sequentially along the evolutionary path leading to the emergence of XBB.1.5. To evaluate the robustness of our claim quantitatively, we performed a sensitivity analysis by reconstructing 10 phylogenetic trees from different sampling datasets with the sampling method mentioned above (**Fig. 1** and **Fig. S1** in the revised manuscript). Eight out of ten trees support the proposed order of mutation occurrences (S:G252V, ORF8:G8stop, and S:F486P). The occurrence of specific mutations was undetectable from the other two trees due to the uncertainty of tree topology, where XBB.2 was positioned within the XBB.1 clade. Furthermore, all the trees support our statement that XBB.1.5 evolved from an ancestor in XBB.1 lineage with Shimodaira-Hasegawa approximate likelihood ratio test (SH-aLRT) and ultrafast bootstrap supporting values of 75.0–90.1 and 31–84, respectively. Together, we concluded that our phylogenetic analyses are reliable and provides substantial insight to our claims.

Supplementary Fig. 1. Sensitivity analysis on sequence down-sampling.

The ten maximum likelihood-based phylogenetic trees of SARS-CoV-2 in the XBB lineage used in the sensitivity analysis. The bottom-right tree is the tree shown in **Fig. 1**. The XBB.1.4.1, XBB.3.1, and XBB.4.1 sublineages are included in the XBB.1.4, XBB.3, and XBB.4 lineages, respectively. Diamonds represent the occurrence of mutations of interest. Only mutation occurrences at internal nodes with at least 20 and also a half of descendant tips harboring the mutation are shown. Numbers at diamonds represent Shimodaira-Hasegawa-like approximate likelihood ratio test and ultrafast bootstrap supporting values, respectively.

3. The ancestral reconstruction of the mutations leading to the XB.1.5 variant is also unreliable, due to sampling bias in the data (se above) and because the method does not take into account evolutionary relationships.

In the original manuscript, we inferred the ancestral mutation pattern using the maximum parsimony method. This approach does not consider branch length nor nucleotide substitution model as pointed out by the reviewer. To address this concern, we employed the ancestral sequence reconstruction method in IQ-TREE 2 (Reference #47), which takes both branch length and nucleotide substitution models into account. We subsequently investigated the order of occurrences of mutations of interest along the ten new phylogenetic trees (**Fig.1** and **Fig. S1** in the revised manuscript). As described above, there are eight from ten trees supporting the proposed order of mutation occurrences (**Lines 151–157**), ensuring that our claim on the order of mutation occurrences is reliable.

Regarding all the three Major remarks, we have revised the related Results, Method, Table legend, and Figure legends sections accordingly. Please find the details below.

Lines 142-156 in the revision:

In this study, we traced the evolutionary history of XBB.1.5 through the reconstruction of a maximum likelihood-based phylogenetic tree using genomic sequences of SARS-CoV-2 isolates in the XBB lineage (**Fig. 1**, **Fig. S1**, and **Table S1**). Regarding the difficulty of SARS-CoV-2 phylogenetic analysis due to low supportive values¹⁸, we reconstructed ten phylogenetic trees in total and compared their topology. All the trees suggest that XBB.1.5 emerged from an ancestor in the XBB.1 lineage. Compared with XBB, XBB.1.5 harbors S:G252V, S:F486P, and ORF8:G8stop mutations. To elucidate the occurrence order of these mutations, we reconstructed the ancestral genomic sequences and investigated the mutation occurrence along the phylogenetic trees (**Fig. 1** and **Fig. S1**). Our results from eight of the ten trees suggest that the S:G252V mutation putatively occurred first in an ancestor of XBB.1. Although most XBB.1 harbor the ORF8:G8stop mutation, this mutation occurred during the diversification of XBB.1 rather than in the most recent common ancestor (MRCA) of XBB.1. The S:S486P mutation occurred later in the putative MRCA of XBB.1.5, followed by the diversification of XBB.1.5, suggesting the contribution of the S:S486P mutation to the increased viral fitness of XBB.1.5 compared with XBB.1⁶.

Lines 464-472 in the revision:

Fig. 1 | Evolutionary history of the XBB.1.5 sublineage

A representative maximum likelihood-based phylogenetic tree of SARS-CoV-2 in the XBB lineage. The XBB.1.4.1, XBB.3.1, and XBB.4.1 sublineages are included in the

XBB.1.4, XBB.3, and XBB.4 lineages, respectively. Diamonds represent the occurrence of mutations of interest. Only mutation occurrences at internal nodes with at least 20 and also a half of descendant tips harboring the mutation are shown. Numbers at diamonds represent Shimodaira-Hasegawa-like approximate likelihood ratio test and ultrafast bootstrap supporting values, respectively.

Lines 646-687 in the revision:

Phylogenetic tree and ancestral genomic sequence reconstruction

We obtained surveillance data of 14,617,387 SARS-CoV-2 isolates from the GISAID database on January 24, 2023 (<https://www.gisaid.org>)⁴³. The PANGO lineage of each SARS-CoV-2 isolate was also assigned using NextClade v2.14.0⁴⁴ in parallel. We excluded the data of any SARS-CoV-2 isolate that i) lacked GISAID and NextClade PANGO lineage information; ii) was isolated from non-human hosts; iii) was sampled from the original passage; and iv) whose genomic sequence was no longer than 28,000 base pairs and contained $\geq 2\%$ of unknown (N) nucleotides. In total, the filtered data contain the data of 29,608 SARS-CoV-2 isolates in XBB lineage. We used the GISAID PANGO lineage classification in downstream analyses.

We performed random sampling to retrieve up to 10 genomic sequences from the combination of each XBB sublineage and each country, resulting in a total of 2,350 genomic sequences of SARS-CoV-2. The sampled genomic sequences were then aligned to the genomic sequence of Wuhan-Hu-1 SARS-CoV-2 isolate (NC_045512.2) with reference-guide multiple pairwise alignment strategy using ViralMSA v1.1.24⁴⁵. Gaps in the alignment were removed automatically using TrimAl v1.4.rev22 with -gappypout mode⁴⁶. A preliminary maximum likelihood-based phylogenetic tree of representative XBB sublineages was reconstructed from the alignment using IQ-TREE v2.2.0⁴⁷. The best-fit nucleotide substitution model was selected automatically using ModelFinder implemented in the IQ-TREE suite⁴⁸. Branch support was assessed using Shimodaira-Hasegawa-like approximate likelihood ratio test and ultrafast bootstrap approximation⁴⁹ with 1,000 replicates. We subsequently removed genomic sequences causing branch length outliers in the preliminary tree determined by Rosner test implemented in the EnvStats R package v2.7.0⁵⁰ using R v4.2.2⁵¹. The final tree was then reconstructed (EPI SET ID: EPI_SET_231124cy) using the methods described earlier. The tree was visualized using ggtree R package v3.6.2⁵². The XBB sublineage was used as an outgroup for tree rooting. The ancestral genomic sequence was reconstructed from the genomic sequences without gap trimming using empirical Bayesian method implemented in the IQ-TREE suite, which consider both branch length

and nucleotide substitution model⁴⁷. The best-fit model used in the ancestral genomic sequence reconstruction was the same model used in the phylogenetic tree reconstruction.

Sensitivity analysis on the effect of genomic sequence down-sampling

We performed the sensitivity analysis to assess the effect of genomic sequence down-sampling on reliability of phylogenetic analyses. Genomic sequences of SARS-CoV-2 were randomly sampled ten times to generate ten different datasets (**Table S1**). We then performed reconstructions of phylogenetic tree and ancestral genomic sequence as described earlier. Topology of the phylogenetic trees and the order of mutation occurrences were subsequently compared.

Lines 1353-1354 in the revision:

Table S1. GISAID Accession ID of SARS-CoV-2 genomic sequences used in phylogenetic tree and ancestral genomic sequence reconstruction.

Lines 1360-1369 in the revision:

Fig. S1. Sensitivity analysis on sequence down-sampling.

The ten maximum likelihood-based phylogenetic trees of SARS-CoV-2 in the XBB lineage used in the sensitivity analysis. The bottom-right tree is the tree shown in **Fig. 1**. The XBB.1.4.1, XBB.3.1, and XBB.4.1 sublineages are included in the XBB.1.4, XBB.3, and XBB.4 lineages, respectively. Diamonds represent the occurrence of mutations of interest. Only mutation occurrences at internal nodes with at least 20 and also a half of descendant tips harboring the mutation are shown. Numbers at diamonds represent Shimodaira-Hasegawa-like approximate likelihood ratio test and ultrafast bootstrap supporting values, respectively.

Reviewers' Comments:

Reviewer #1:

Remarks to the Author:

The authors answered to all raised comments and suggestions and integrated brief explanations into the manuscript.

Although the majority of points was covered well, I think the manuscript would be improved if the following discussed aspect:

"Figure 5B: You describe the enhanced transmissibility of XBB.1.5, but in oral swabs even less RNA load is detected at 5dpi compared to Delta and XBB.1. How do you explain that?

We have two reasons to conclude that XBB.1.5 might have acquisition of enhanced transmissibility. First, the Spike-hACE2 complex of XBB.1 and XBB.1.5 showed that binding affinity is enhanced by S486P mutation in XBB.1.5 as discussed above. Secondly, in hamsters, XBB.1.5 possesses slightly higher ability of viral shedding and dissemination compared with the ancestral XBB.1. However, because XBB.1.5 acquired the mutation to disrupt the function of ORF8 and did not suppress MHC-1 efficiently, the virus shedding impaired at the later time point."

would be briefly integrated into the discussion (if not already done).

Thanks to the authors for the work to cover all raised points and therefore improve the manuscript.

Reviewer #2:

Remarks to the Author:

Authors have addressed many of the comments raised during initial peer review to my satisfaction. Inclusion of infectious viral titers for both in vitro and in vivo studies, now in the supplemental material, strengthens the study in particular, as does toning down the speculative text regarding transmissibility implications of these findings. However, these authors continue to make statements brought up during initial peer review by multiple reviewers which still appear unsubstantiated by the data presented.

This is most apparent with in vivo results from hamster studies (Figure 5). Lines 299-300, authors state "...viral dissemination of XBB.1.5 in the lungs is slightly higher than that of the ancestral XBB.1" but there was no statistically significant difference between lung viral titers (new data in Supplemental Figure 4A), and the viral RNA results shown in Figure 5B, which technically being statistically significant with the tests employed, do not appear biologically meaningful (the authors report this data using a log scale which only extends to the highest and lowest titers reported to make the results seem more different and spaced apart, but if these data were reported on a scale that started at 0 or 1 the differences depicted would look very small). The histopathological data reported with scores in Figure 5E and Supplemental Figure 4B similarly does not appear to present striking and biologically meaningful differences between XBB.1 and XBB.1.5, yet authors emphasize this difference in the discussion on lines 332-3 and 436-7. Thus, with viral RNA/titer readouts and histopathological assessments which offer generally mild and modest differences, authors are still concluding "lowered intrinsic pathogenicity" with XBB.1.5 (line 438) and in the abstract (lines 92-3). As the two mutations introduced into these viruses did not significantly alter viral RNA readouts compared to either XBB.1 or XBB.1.5 viruses (Figure 7B-C) and histopathological assessments in the second round of hamster studies fail to support striking differences in pathogenicity between these strains, it is unclear to me what data the authors are interpreting to draw these conclusions.

Furthermore, quantification of plaque size differences between XBB.1 and XBB.1.5 from plaque assays (while interesting) is not a standard readout to quantify cytopathetic effect, and as such the statement on lines 252-3 that XBB.1.5 is associated with higher CPE relative to XBB.1 still appears to be unsubstantiated. Furthermore, the text overall is lacking in harmonization between reporting that XBB.1.5 is associated with more cellular damage post-infection than XBB.1 while

concurrently pushing that XBB.1 has heightened pathogenicity in vivo than XBB.1.5.

In my eyes, the differences between these two strains appear generally negligible between both experimental approaches. As such, the statement in the abstract (lines 96-7) that their study is identifying two mutations that are “involved in the difference of viral pathogenicity” between these two variants of concern (lines 371-2) is very challenging to interpret, as the differences between the two VOCs are so few to begin with.

Reviewer #4:

Remarks to the Author:

The authors have done an excellent job in addressing every issue raised by the reviewers of the previous manuscript. I have no further comments.

Answers to Reviewer #1

The authors answered to all raised comments and suggestions and integrated brief explanations into the manuscript.

Although the majority of points was covered well, I think the manuscript would be improved if the following discussed aspect:

"**Figure 5B**: You describe the enhanced transmissibility of XBB.1.5, but in oral swabs even less RNA load is detected at 5dpi compared to Delta and XBB.1. How do you explain that?

We have two reasons to conclude that XBB.1.5 might have acquisition of enhanced transmissibility. First, the Spike-hACE2 complex of XBB.1 and XBB.1.5 showed that binding affinity is enhanced by S486P mutation in XBB.1.5 as discussed above. Secondly, in hamsters, XBB.1.5 possesses slightly higher ability of viral shedding and dissemination compared with the ancestral XBB.1. However, because XBB.1.5 acquired the mutation to disrupt the function of ORF8 and did not suppress MHC-1 efficiently, the virus shedding impaired at the later time point."

would be briefly integrated into the discussion (if not already done).

Thank you again for a time to evaluate our manuscript. We included the statements as you suggested in this revision accordingly.

Lines 449-450 in the revision:

Although XBB1.5 might have acquisition of enhanced transmissibility, these data...

Lines 465-472 in the revision:

We have two reasons to conclude that XBB.1.5 might have acquisition of enhanced transmissibility. First, the Spike-hACE2 complex of XBB.1 and XBB.1.5 showed that binding affinity is enhanced by S486P mutation in XBB.1.5 as discussed above. Secondly, in hamsters, XBB.1.5 possesses slightly higher ability of viral shedding and dissemination compared with the ancestral XBB.1. However, because XBB.1.5 acquired the mutation to disrupt the function of ORF8 and did not suppress MHC-1 efficiently, the virus shedding impaired at the later time point (**Fig. 5B**). Moreover, our series...

Answers to Reviewer #2

Authors have addressed many of the comments raised during initial peer review to my satisfaction. Inclusion of infectious viral titers for both in vitro and in vivo studies, now in the supplemental material, strengthens the study in particular, as does toning down the speculative text regarding transmissibility implications of these findings. However, these authors continue to make statements brought up during initial peer review by multiple reviewers which still appear unsubstantiated by the data presented.

Thank you again for giving us an opportunity to improve our manuscript. Please kindly find the comment below.

This is most apparent with in vivo results from hamster studies (**Figure 5**). Lines 299-300, authors state "...viral dissemination of XBB.1.5 in the lungs is slightly higher than that of the ancestral XBB.1" but there was no statistically significant difference between lung viral titers (new data in **Supplemental Figure 4A**), and the viral RNA results shown in **Figure 5B**, which technically being statistically significant with the tests employed, do not appear biologically meaningful (the authors report this data using a log scale which only extends to the highest and lowest titers reported to make the results seem more different and spaced apart, but if these data were reported on a scale that started at 0 or 1 the differences depicted would look very small). The histopathological data reported with scores in **Figure 5E** and **Supplemental Figure 4B** similarly does not appear to present striking and biologically meaningful differences between XBB.1 and XBB.1.5, yet authors emphasize this difference in the discussion on lines 332-3 and 436-7. Thus, with viral RNA/titer readouts and histopathological assessments which offer generally mild and modest differences, authors are still concluding "lowered intrinsic pathogenicity" with XBB.1.5 (line 438) and in the abstract (lines 92-3). As the two mutations introduced into these viruses did not significantly alter viral RNA readouts compared to either XBB.1 or XBB.1.5 viruses (**Figure 7B-C**) and histopathological assessments in the second round of hamster studies fail to support striking differences in pathogenicity between these strains, it is unclear to me what data the authors are interpreting to draw these conclusions.

Furthermore, quantification of plaque size differences between XBB.1 and XBB.1.5 from plaque assays (while interesting) is not a standard readout to quantify cytopathic effect, and as such the statement on lines 252-3 that XBB.1.5 is associated with higher CPE relative to XBB.1 still appears to be unsubstantiated. Furthermore, the text overall is lacking in harmonization between reporting that XBB.1.5 is associated with more cellular damage post-infection than XBB.1 while concurrently pushing that XBB.1 has heightened pathogenicity in

vivo than XBB.1.5.

In my eyes, the differences between these two strains appear generally negligible between both experimental approaches. As such, the statement in the abstract (lines 96-7) that their study is identifying two mutations that are “involved in the difference of viral pathogenicity” between these two variants of concern (lines 371-2) is very challenging to interpret, as the differences between the two VOCs are so few to begin with.

As request from the reviewer, we removed all of the interruption regarding differences of viral pathogenicity between XBB.1 and XBB.1.5 and revised the sentences describing data sets accordingly. As per cytopathic effect, we further provided the experimental data using Viral ToxGlo assay (Promega) to measure cellular ATP upon infection with Delta, XBB.1, and XBB.1.5. As shown in **Supplementary Figure 3F**, a significantly higher cytopathic effects were induced by infection with XBB.1.5 infection than those by XBB.1 in Vero cells, strengthen the data of plaque assay in **Supplementary Figure 3E**. Together with the data of the *in vivo* experiments, XBB.1.5 exhibited higher cytotoxicity than XBB.1 and might provoke enhanced immune response in vivo, leading to viral attention. But, as you pointed out, we don't overstate in this revision.

Lines 95-96 in the revision:

We provide the intrinsic pathogenicity of XBB.1 and XBB.1.5 in hamsters.

Lines 265-267 in the revision:

This finding was further supported by the Viral ToxGlo assay where we observed a significantly higher cytopathic effects induced by XBB.1.5 than those induced by XBB.1 in Vero cells (**Supplementary Fig. 3F**).

Lines 382-383 in the revision:

...ORF8:G8stop alter viral function and are involved in viral pathogenicity in hamsters.

Lines 446-448 in the revision:

On histopathological analysis, the severity of bronchitis/bronchiolitis was diminished upon infection with XBB.1.5...

Lines 450-451 in the revision:

...the acquisition of the S:S486P and ORF8 stop mutations are involved in the pathogenicity of XBB.1.5.

Lines 671-676 in the revision:

Viral ToxGlo assay

Vero cells or VeroE6/TMPRSS2 cells (18,000 cells per well) were seeded into a 96-well plate. After overnight incubation, cells were infected with SARS-CoV-2 (90, or 900 TCID₅₀). At 72 h.p.i., the viral-induced cytopathic effects (CPE) were quantified with Viral ToxGlo Assay (Promega, Cat# G8942) according to the manufacturer's protocol.

Answers to Reviewer #4

The authors have done an excellent job in addressing every issue raised by the reviewers of the previous manuscript. I have no further comments.

Thank you very much for spare your time to evaluate our manuscript. I'm glad to hear your high evaluation toward the previous revision.